# DiMeR: Disentangled Mesh Reconstruction Model with Normal-only Geometry Training

**Lutao Jiang**[1*] **Jiantao Lin**[1*] **Kanghao Chen**[1*] **Wenhang Ge**[1*] **Xin Yang**[1,2] **Yifan Jiang**[1]
**Yuanhuiyi Lyu**[1] **Xu Zheng**[1,3] **Jing Li**[5] **Yinchuan Li**[4] **Ying-Cong Chen**[1,2†]
[1]The Hong Kong University of Science and Technology (Guangzhou)
[2]The Hong Kong University of Science and Technology
[3]INSAIT, Sofia University "St. Kliment Ohridski"  [4]Knowin  [5]ByteDance

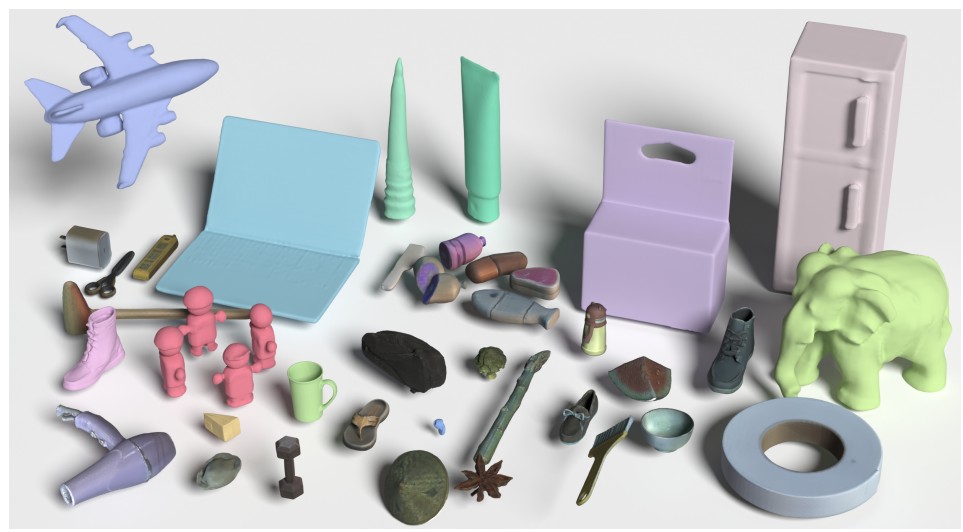

Figure 1: DiMeR takes the image or text inputs and generates detailed 3D meshes.

## ABSTRACT

We propose DiMeR, a novel geometry-texture disentangled feed-forward model with 3D supervision for sparse-view mesh reconstruction. Existing methods confront two persistent obstacles: (i) textures can conceal geometric errors, *i.e.*, visually plausible images can be rendered even with wrong geometry, producing multiple ambiguous optimization objectives in geometry-texture mixed solution space for similar objects; and (ii) prevailing mesh extraction methods are redundant, unstable, and lack 3D supervision. To solve these challenges, we rethink the inductive bias for mesh reconstruction. First, we disentangle the unified geometry-texture solution space, where a single input admits multiple feasible solutions, into geometry and texture spaces individually. Specifically, given that normal maps are strictly consistent with geometry and accurately capture surface variations, the **normal maps serve as the only input** for geometry prediction in DiMeR, while the texture is estimated from RGB images. Second, we streamline the algorithm of mesh extraction by eliminating modules with low performance/cost ratios and redesigning regularization losses with 3D supervision. Notably, DiMeR still accepts raw RGB images as input by leveraging foundation models for normal prediction. Extensive experiments demonstrate that DiMeR generalises across sparse-views-to-3D, single-image-to-3D, and text-to-3D tasks, consistently outperforming baselines. On the GSO and OmniObject3D datasets, DiMeR significantly reduces Chamfer Distance by more than **30%**. Project Page.

## 1 INTRODUCTION

The tasks of 3D reconstruction and generation have garnered significant attention, largely due to the advancements made by NeRF (Mildenhall et al., 2021) and 3DGS (Kerbl et al., 2023). However,

---

* Contribute equally.  † Corresponding author.

transforming them into the mesh poses a challenge. In this paper, we focus on mesh representation, which is easy to adapt to downstream applications, such as the gaming industry, VR, robotics, *etc*.

Enhanced by the introduction of the extensive 3D dataset, Objaverse (Deitke et al., 2023; 2024), numerous 3D reconstruction and generative models emerge. One notable advancement is LRM (Hong et al., 2023), which pioneers the feed-forward generation of a NeRF model from RGB images. Subsequent works (Xu et al., 2024a; Wei et al., 2024; Wang et al., 2025; Yang et al., 2024; Liu et al., 2024; Ge et al., 2024) extend LRM's NeRF representation to mesh. However, two key issues persist in these methods. **First,** the reliance on RGB images as input leads to significant ambiguity in training. As shown in Fig. 2(a), the texture often hides the underlying geometry, thereby leading to the conflict of training objectives exhibited in Fig. 2(b). Furthermore, as demonstrated in Fig. 2(c), RGB images can be rendered from compositions of multiple wrong geometries and textures, driving the network toward an undesirable averaged solution. **Second**, most of the existing mesh reconstruction methods employ FlexiCubes (Shen et al., 2023) to extract the mesh and utilize differential rasterization for optimization. However, the Signed Distance Field (SDF) grid defined in FlexiCubes only promises the meaning of positive and negative signs for surface extraction, which makes it difficult to apply 3D supervision. Moreover, some of its components are redundant for this task, and its regularization losses lead to serious instability in training.

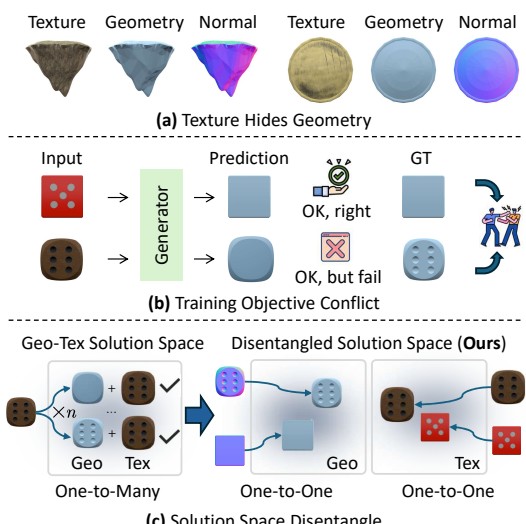

Figure 2: (a) exhibits difficulty in distinguishing geometry from RGB images. (b) shows the conflict input-GT pairs in datasets due to problem (a), hindering the training. (c) illustrates our idea: disentangle the mixed solution space containing multiple feasible solutions into two separate spaces with unbiased input. Samples are from the Objavers dataset (Deitke et al., 2023).

To solve these two challenges, we propose DiMeR, a geometry-texture disentangled feed-forward sparse-view mesh reconstruction model with 3D supervision. To address the first challenge, training ambiguity, we exploit the inductive bias derived from the **consistency between normal maps and 3D geometry**. As shown in Fig. 2 (a) and (c), the normal maps consistently align with the surface of the 3D model, offering a more reliable input format for geometry reconstruction. Building on this inductive bias and the Principle of Occam's Razor (Blumer et al., 1987), we disentangle the geometry-texture unified solution space, where multiple solutions correspond to one input, into two individual spaces. To get the geometry, we **exclusively** utilize normal maps as the sole input, while utilizing RGB images for obtaining the texture. Each of them is trained with task-specific supervision: the geometry branch is constrained by normal, depth, and silhouette-mask losses, whereas the texture branch is guided by appearance-based objectives. Moreover, an accurate geometry should be capable of rendering the light map correctly under arbitrary environmental conditions and various materials. Therefore, we add the statistical expectation supervision signal by placing the predicted untextured mesh model in multiple environments with randomly assigned materials. To address the second challenge, limitations remaining in FlexiCubes, we replace the original regularization losses with eikonal loss (Gropp et al., 2020) and incorporate 3D supervision via ground-truth mesh. Furthermore, we simplify the modules with low performance/cost ratios. Our improved mesh extraction algorithm allows us a higher extraction resolution, compared with other reconstruction models.

Leveraging recent foundation models for normal map prediction (Ye et al., 2024; He et al., 2024), we can generate highly accurate normal maps from RGB images with minimal error and latency (**only 200ms**). To validate this point and choose the optimal model for DiMeR, we conduct a benchmark evaluation for object-level normal prediction. To further improve the robustness of DiMeR in practical applications, we introduce noise to the input, normal maps, during training. **Equipped with these models, DiMeR also accepts RGB images as raw input, the same as other methods.** Our DiMeR model is capable of effectively handling various tasks, including sparse-view reconstruction, single-image-to-3D, and text-to-3D. Extensive experiments demonstrate that DiMeR sig-

nificantly outperforms previous methods. Specifically, on the GSO dataset, DiMeR reduces Chamfer Distance by **22%**, with an upper bound improvement of **32%** when using real normal map inputs. In general, our contributions can be summarized as follows:

- Rethinking the inductive bias for mesh reconstruction, we propose DiMeR, a disentangled framework to train and predict geometry from normal maps and texture from RGB images separately, with decoupled supervision signals.

- We enhance the mesh extraction algorithm for this task, introduce the 3D GT supervision, and Physics-based Rendering Expectation losses.

- Numerous experiments demonstrate the superiority and robustness of our DiMeR on reconstruction, single-image-to-3D, and text-to-3D tasks. We also conduct a benchmark for the foundation models of normal map prediction in object-level tasks.

## 2 RELATED WORKS

### 2.1 3D GENERATIVE MODELS

Building upon advancements in 2D diffusion models, DreamFusion (Poole et al., 2022) introduced score distillation sampling (SDS) to train 3D representation models like NeRF (Mildenhall et al., 2021) and 3DGS (Kerbl et al., 2023) based on text input. Subsequently, numerous methods have been developed to enhance this approach (Wang et al., 2023b; Liang et al., 2023; Chen et al., 2023; Shi et al., 2023b; Wang et al., 2023a; Metzer et al., 2023; Bai et al., 2023; Jiang et al., 2024; Li et al., 2025; Tang et al., 2023). However, a significant limitation of these methods is the need to train a separate 3D model for each text input, which can take tens of minutes or even hours per text. Some approaches attempt to address this by employing SDS to train a feed-forward network (Lorraine et al., 2023; Jiang & Wang, 2024; Li et al., 2023b; Qian et al., 2024), but these are limited to a few specific text subjects, reducing the diversity of the outputs. Recently, the introduction of large-scale 3D datasets, such as Objaverse (Deitke et al., 2023; 2024), has enabled models like LRMs (Hong et al., 2023; Tochilkin et al., 2024) to explore feed-forward reconstruction from a single image. Following this, several methods have been developed to create sparse-view reconstruction models (Tang et al., 2025; Xu et al., 2024b; Zhang et al., 2024a; Li et al., 2023a) based on NeRF or 3DGS. To support real-world applications, leveraging differential marching cube algorithms (Wei et al., 2023; Shen et al., 2023), some methods focus on direct mesh generation (Xu et al., 2024a; Wei et al., 2024; Wang et al., 2025; Liu et al., 2024; Ge et al., 2024). Additionally, several 3D diffusion models (Zhang et al., 2023; Li et al., 2024b; Zhang et al., 2024b; Ren et al., 2024; Zhang & Wonka, 2024; Xiang et al., 2024; Hua et al., 2025; Jia et al., 2025) emerge, but they are limited to the generation task and lack strict correspondence with input images. Moreover, their inference latency ranges from tens of seconds to several minutes. Inspired by auto-regressive models (Tian et al., 2024; Zhou et al., 2024), some researchers have shifted focus to mesh AR generation (Siddiqui et al., 2024; Chen et al., 2024b;a; Tang et al., 2024). However, these methods typically require the number of mesh faces to be fewer than 6,000. Concurrently, similar to us, Hi3DGen (Ye et al., 2025) also found that exclusive utilization of normal maps can enhance the quality of geometry and implemented a diffusion model based on this.

In this paper, we focus on feed-forward sparse-view mesh reconstruction. Differently, we disentangle the framework into dual branches that predict geometry solely from normal and predict texture from RGB. To ensure that each branch performs its intended role, we assign branch-specific, unambiguous supervision signals.

### 2.2 MULTI-VIEW DIFFUSION MODEL

Multi-view diffusion models are designed to generate multi-view images or normal maps from a single image or text prompts, instead of directly producing corresponding 3D models. This approach is gaining popularity due to the relative simplicity of its task definition, where multi-view images are synthesized first, followed by the use of sparse-view reconstruction models to complete the 3D model generation process. Zero123 (Liu et al., 2023) introduces explicit control by embedding camera parameters into the conditions of 2D diffusion models. Following, many methods have achieved significant success to synthesis multi-view images and normal maps (Shi et al., 2023a;b; Li et al., 2023a; Melas-Kyriazi et al., 2024; Wang & Shi, 2023; Voleti et al., 2024; Wu et al., 2024; Li et al., 2024a; Long et al., 2024; Lu et al., 2024; Lin et al., 2025). We employ the image-input 2.5D model, such as zero123++ (Shi et al., 2023a) and Era3D (Li et al., 2024a), to perform the single-image-to-3D task, while we use the text-input 2.5D diffusion model, such as Kiss3DGen,

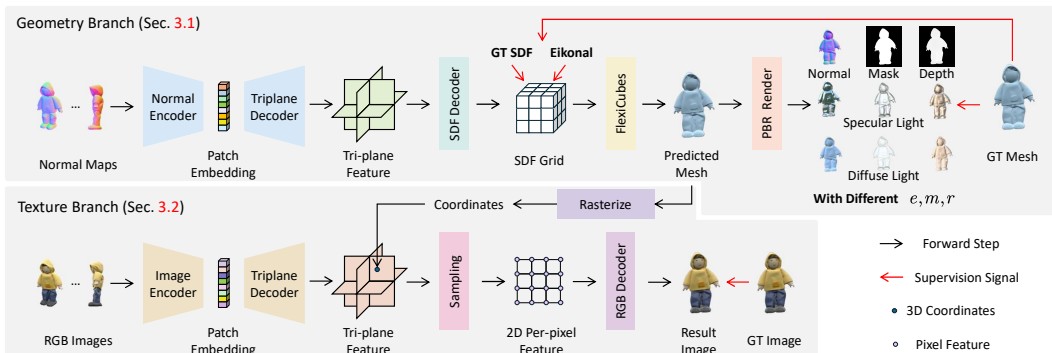

Figure 3: The framework of our DiMeR. The upper part is the geometry branch, and **exclusively** uses normal maps as input. The lower part is the texture branch.

to accomplish the text-to-3D task. With the continuing progress of such models, DiMeR has the potential to further enhance generation quality.

### 2.3 NORMAL PREDICTION FOUNDATION MODELS

Surface normals precisely describe local surface variation and orientation, making them crucial for 3D reconstruction. The robustness and accuracy of recent normal prediction foundation models have reached practical levels. Marigold (Ke et al., 2024) first integrates diffusion models into depth and normal estimation, by preserving the prior that the visual generative models learned. Subsequent works (Fu et al., 2024; Bae & Davison, 2024; Ye et al., 2024; He et al., 2024) have further boosted performance while markedly reducing inference latency. Collectively, these advances provide the possibility for high-quality 3D reconstruction exclusively from predicted normal maps, enabling DiMeR with the RGB image as input.

## 3 METHOD

As shown in Fig. 3, the objective of our DiMeR is to reconstruct the 3D mesh geometry from normal maps and derive texture from RGB images. We introduce Geometry Branch in Sec. 3.1, Texture Branch in Sec. 3.2, and applications in Sec. 3.3.

### 3.1 GEOMETRY BRANCH

As illustrated in Fig. 2, a single RGB image admits many equally plausible solutions in the geometry–texture joint solution space, encouraging the network to learn over-smoothed averages. Normal maps, in contrast, are uniquely determined by the underlying surface and faithfully encode fine geometric variation. Guided by Occam's razor (Blumer et al., 1987), we therefore feed only normal maps to the geometry branch, eliminating appearance-induced ambiguities and simplifying the correspondence between input and output. This design establishes a clearer relationship between the network's inputs and outputs, ultimately reducing the training complexity. Supervision is likewise restricted to geometry-specific losses, discarding ambiguous RGB rendering terms. We further regularize geometry by rendering the untextured mesh with physically based rendering (PBR) under diverse illuminations and materials, matching the resulting lighting maps to statistical expectations. Finally, we improve the mesh extraction algorithm for greater efficiency and robustness and incorporate direct 3-D supervision.

**Network Structure.** As shown in Fig. 3, the geometry branch of our DiMeR model initiates with normal maps $\mathcal{N} \in \mathbb{R}^{K \times H \times W \times 3}$ of $K$ randomly selected views, alongside their associated camera embeddings $\zeta \in \mathbb{R}^{K \times 16}$. We opt for a random sampling of input views to improve the model's capability to interpret camera embeddings from arbitrary directions and add slight noise to them, thereby enhancing robustness and reducing dependency on specific input configurations. Furthermore, this also reduces the requirements for the user input, allowing users to provide inputs from unfixed view directions. The normal maps $\mathcal{N}$ and their associated camera embeddings $\zeta$ are encoded into patch-wise representations $\mathcal{P}_g \in \mathbb{R}^{K \times D \times C}$ using a ViT-based Normal Encoder, where $D$ is the number of patches of each view and $C$ is the dimension of the feature. Similar to the approach taken by LRM (Wei et al., 2024), we utilize a Triplane Decoder to gather information from the Patch Embedding $\mathcal{P}_g$ using several transformer layers (Vaswani et al., 2017) to synthesize triplane (Chan et al., 2022) features $\mathcal{F}_g \in \mathbb{R}^{3 \times H' \times W' \times C_g}$. Subsequently, we extract an SDF grid from the triplane fea-

tures $\mathcal{F}_g$ to apply the differential isosurface construction algorithm, FlexiCubes (Shen et al., 2023), to obtain the vertices and faces for the mesh. Finally, we can rasterize the mesh to get the normal maps, masks, and depth maps for arbitrary views. By providing the environment map and assigning different materials (metallic and roughness) to the mesh, we can render the light map (including specular and diffuse) using PBR, for enhancing the supervision from different lighting conditions, which will be introduced in the following part.

**Mesh Extraction Algorithm.** Original FlexiCubes requires two MLP networks to output different weights (each edge and vertex in the grid) and the deformation of the grid. However, this incurs excessive computational and GPU memory overhead. Specifically, for a $N^3$ grid, it needs to compute the deformation of $N^3$ vertices and the weight of $12 \times N^3$ edges and $8 \times N^3$ vertices. Though it is powerful for the tasks of Flexicubes itself, extensive experiments prove that these components contribute disproportionately high computational overhead with minimal performance gains. As shown in Tab. 5, we found that removing these networks from the pre-trained model does not adversely affect performance. Therefore, to enable higher efficient training and higher spatial resolutions, we prune these components to improve computational efficiency and improve the spatial resolution.

**Optimization.** Given the inherent ambiguity introduced by the RGB texture shown in Fig. 2, we exclude RGB loss to enhance training stability. Consequently, we now exclusively employ geometry-related losses to supervise the geometry branch of our model.

In its original implementation, FlexiCubes incorporates three regularization losses to regularize the SDF grid values generated by the network. However, extensive experimentation reveals that this approach yielded low stability (Xu et al., 2024a; Ge et al., 2024). Furthermore, the design does not produce true SDF representations. To address these issues, we employ the eikonal loss (Gropp et al., 2020) to regularize the whole space as the SDF field, specifically by ensuring the norm of the gradient with respect to the coordinates is normalized to 1. Nevertheless, computing the derivative for a $N^3$ grid poses significant challenges in terms of computational and GPU memory costs and potential overfitting at specific grid positions. To mitigate this, we propose randomly sampling positions within the space to compute the eikonal expectation loss, effectively reducing computational demands while maintaining the integrity of the regularization, *i.e.*,

$$\mathcal{L}_{eik} = \mathbb{E}_{\boldsymbol{x}}(\|\nabla_{\boldsymbol{x}}\text{SDF}(\boldsymbol{x})\|_2 - 1)^2, \boldsymbol{x} \in \mathbb{R}^3 \sim Uniform(-1, 1), \tag{1}$$

where we sample $200K$ $\boldsymbol{x}$ in each iteration to calculate the expectation. Moreover, we use the GT SDF value to supervise the SDF value of grid vertices $\boldsymbol{v} \in \mathbb{R}^{N^3 \times 3}$ in FlexiCubes,

$$\mathcal{L}_{sdf} = \|\text{SDF}(\boldsymbol{v}) - \text{SDF}_{\text{GT}}(\boldsymbol{v}))\|_2^2. \tag{2}$$

To reduce computational overhead, we cache these SDF values for each object in the training set.

Drawing inspiration from Photometric Stereo (Woodham, 1980), we introduce the PBR (Kajiya, 1986) losses. The premise is that if the specular and diffuse light maps of a 3D mesh under different environmental lighting conditions and various materials can be accurately rendered in PBR, then the geometry of the predicted mesh model can be deemed correct. Therefore, we introduce the statistical expectation loss of PBR to supervise the geometry branch,

$$\mathcal{L}_{spec} = \mathbb{E}_{e,m,r} \left( \text{Spec}(\hat{\mathcal{O}}, e, m, r) - \text{Spec}(\mathcal{O}, e, m, r) \right)^2$$
$$+ \text{LPIPS} \left( \text{Spec}(\hat{\mathcal{O}}, e, m, r), \text{Spec}(\mathcal{O}, e, m, r) \right), \tag{3}$$

$$\mathcal{L}_{diff} = \mathbb{E}_{e,m,r} \left( \text{Diff}(\hat{\mathcal{O}}, e, m, r) - \text{Diff}(\mathcal{O}, e, m, r) \right)^2$$
$$+ \text{LPIPS} \left( \text{Diff}(\hat{\mathcal{O}}, e, m, r), \text{Diff}(\mathcal{O}, e, m, r) \right), \tag{4}$$

where $\hat{\mathcal{O}}$ and $\mathcal{O}$ are the predicted and ground truth mesh separately, $e$, $m$, $r$ are the randomly sampled environment, metallic, and roughness, $\text{Spec}(\cdot)$ and $\text{Diff}(\cdot)$ are rendering functions of specular and diffuse light map, $\text{LPIPS}(\cdot)$ is the perception loss (Zhang et al., 2018). Notably, during the training, we sample different environment, metallic, and roughness to render the light maps for a single object.

We also employ the commonly used normal, depth, and mask losses to supervise the geometry branch. Specifically,

$$\mathcal{L}_{nor} = \mathcal{M}_{\text{GT}} \otimes (1 - \hat{\mathcal{N}} \cdot \mathcal{N}_{\text{GT}}), \tag{5}$$

$$\mathcal{L}_{dep} = \mathcal{M}_{\text{GT}} \otimes |\hat{\mathcal{D}} - \mathcal{D}_{\text{GT}}|, \tag{6}$$

$$\mathcal{L}_{mask} = (\hat{\mathcal{M}} - \mathcal{M}_{\text{GT}})^2, \tag{7}$$

where $\otimes$ denotes element-wise production, $\mathcal{M}_{\text{GT}}$ and $\hat{\mathcal{M}}$ are the rendered mask from ground truth mesh model and predicted mesh model, similarly, $\mathcal{N}$ and $\mathcal{D}$ represent normal map and depth map. In general, the overall loss function is

$$\mathcal{L}_g = \mathcal{L}_{eik} + \mathcal{L}_{sdf} + \mathcal{L}_{spec} + \mathcal{L}_{diff} + \mathcal{L}_{nor} + \mathcal{L}_{dep} + \mathcal{L}_{mask}. \tag{8}$$

## 3.2 TEXTURE BRANCH

**Network Structure.** As demonstrated in Fig. 3, the texture branch starts from RGB images $\mathcal{I} \in \mathbb{R}^{K \times H \times W \times 3}$ with the camera embeddings $\zeta$. Similar as the geometry branch, we use a ViT-based Image Encoder to get the Patch Embedding $\mathcal{P}_c \in \mathbb{R}^{K \times D \times C}$ and utilize a Triplane Decoder to assemble the information from $\mathcal{P}_c$ to get the triplane features $\mathcal{F}_c \in \mathbb{R}^{3 \times H' \times W' \times C_c}$ for the texture field representation (Oechsle et al., 2019). Given the predicted shape from the geometry branch, we rasterize the vertex coordinates $v$ into image space,

$$Coord_{\mathcal{I}} = \text{Rast}(v, \text{Camera}), \tag{9}$$

where the pixel value of $Coord_{\mathcal{I}} \in \mathbb{R}^{H \times W \times 3}$ is the global coordinate. Next, we query the texture feature $\mathcal{F}_{\mathcal{I}} \in \mathbb{R}^{H \times W \times C}$ on triplane $\mathcal{F}_c$ for each pixel,

$$\mathcal{F}_{\mathcal{I}} = \text{Sample}(Coord_{\mathcal{I}}, \mathcal{F}_c). \tag{10}$$

Finally, we decode the color feature to predict the image

$$\hat{\mathcal{I}} = \text{RGB\_Decoder}(\mathcal{F}_{\mathcal{I}}). \tag{11}$$

**Optimization.** In this branch, we only use RGB loss to supervise the network. Specifically,

$$\mathcal{L}_t = (\hat{\mathcal{I}} - \mathcal{I}_{\text{GT}})^2 + \text{LPIPS}(\hat{\mathcal{I}}, \mathcal{I}_{\text{GT}}). \tag{12}$$

## 3.3 APPLICATIONS

Besides the sparse-view reconstruction task, DiMeR is also capable of performing image/text-to-3D tasks.

**Single-image-to-3D.** Given the input image, we first employ Zero-1-2-3++ (Shi et al., 2023a) or Era3D (Li et al., 2024a) to generate six images from different viewpoints. Specifically, the output from zero123++ consists of six views, including the combinations of azimuth and elevation, $(30, 20)$, $(90, -10)$, $(150, 20)$, $(210, -10)$, $(270, 20)$, and $(330, -10)$. Next, we apply the SoTA normal prediction model Lotus (He et al., 2024) or StableNormal (Ye et al., 2024) to predict the normal maps for these six views. Since the predicted normal maps are initially in the local camera coordinate system, we subsequently transform them into the global coordinate system using the transformation matrices corresponding to the six view directions. Finally, we feed the six transformed normal maps and the RGB images into our DiMeR model to generate the textured mesh.

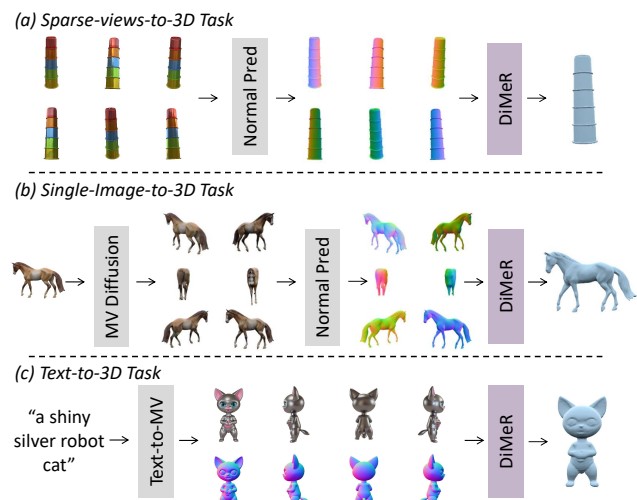

Figure 4: Pipelines for sparse-views, single-image-, and text-to-3D.

**Text-to-3D.** This task is approached through two distinct pipelines: (I) The first pipeline involves using a text-to-image model to generate an RGB image from the input text. Subsequently, we apply the single-image-to-3D pipeline to complete the reconstruction. (II) With the advancement of diffusion models, Kiss3DGen (Lin et al., 2025) fine-tunes the SoTA text-to-image generative model, FLUX (BlackForestLabs, 2024), to simultaneously output RGB images along with corresponding normal maps, ensuring high multi-view consistency. Since our DiMeR supports a dynamic number of input views, we can directly feed the four views from Kiss3DGen into DiMeR for 3D model reconstruction. The generated high-quality models are presented in Fig. 1 and the supplementary materials.

Table 1: Quantitative results for reconstruction task. CD means Chamfer Distance. DiMeR (Lotus) and DiMeR (SN) are the reconstruction results from the normal map predicted by Lotus (He et al., 2024) and StableNormal (Ye et al., 2024) separately. DiMeR (GT) is from the ground truth normal. value means first-best, value means second-best, value means third-best.

| Dataset | GSO | | | | | OmniObject3D | | | | |
|---|---|---|---|---|---|---|---|---|---|---|
| Metric | CD (↓) | F1 (↑) | PSNR (↑) | SSIM (↑) | LPIPS (↓) | CD (↓) | F1 (↑) | PSNR (↑) | SSIM (↑) | LPIPS (↓) |
| InstantMesh | 0.045 | 0.964 | 18.51 | 0.846 | 0.150 | 0.039 | 0.983 | 18.44 | 0.842 | 0.153 |
| PRM | 0.041 | 0.977 | 21.68 | 0.869 | 0.126 | 0.034 | 0.991 | 21.65 | 0.865 | 0.135 |
| DiMeR (GT) | 0.028 | 0.992 | 23.40 | 0.883 | 0.095 | 0.024 | 0.996 | 23.04 | 0.871 | 0.112 |
| Δ | 31.7% ↓ | +0.015 | +1.72 | +0.014 | 24.6% ↓ | 29.4% ↓ | +0.005 | +1.39 | +0.006 | 17.0% ↓ |
| DiMeR (Lotus) | 0.033 | 0.988 | 22.57 | 0.874 | 0.103 | 0.034 | 0.989 | 21.88 | 0.866 | 0.126 |
| DiMeR (SN) | 0.032 | 0.988 | 22.89 | 0.875 | 0.103 | 0.030 | 0.993 | 22.15 | 0.865 | 0.121 |

## 4 EXPERIMENT

### 4.1 IMPLEMENTATION DETAILS

We train DiMeR with the filtered Objaverse (Deitke et al., 2023) according to the mesh quality, in a total of $98,526$ objects. For test datasets, we choose the widely used GSO (Downs et al., 2022) and OmniObject3D (Wu et al., 2023). We use all $1,029$ objects in GSO and randomly select 5 objects for each class in OmniObject3D. More details can be found in appendix.

### 4.2 QUANTITATIVE COMPARISON

**Reconstruction Task.** As shown in Tab. 1, we compare our DiMeR on the sparse-view reconstruction task using the same 6 randomly sampled input views. Since some sparse-view reconstruction methods, like CRM (Wang et al., 2025), are limited to only support specific views (six orthogonal views), we compare them in single-image-to-3D tasks. Additionally, because MeshFormer (Liu et al., 2024) is not open-source work, we are unable to perform an accurate quantitative comparison. Therefore, we only provide qualitative visual comparisons. For the comparison, we select state-of-the-art (SoTA) methods that are accessible, including InstantMesh (Xu et al., 2024a) and PRM (Ge et al., 2024). Experiments show that our method can surpass the SoTA methods by a large margin, whatever using GT (31.7% gain) or predicted normal maps (22.0% gain) from StableNormal-Turbo (Ye et al., 2024). **Notably, when equipped with normal map prediction models, the input to DiMeR remains the same as the baselines, relying solely on RGB images**. Furthermore, following the improvement of normal prediction models, there is still room for continued improvement in the performance of DiMeR.

**Sinle-Image-to-3D Task.** As demonstrated in Tab. 2, we compare our DiMeR with CRM (Wang et al., 2025), MeshLRM (Wei et al., 2024), InstantMesh (Xu et al., 2024a), and PRM (Ge et al., 2024), using same single image input. Our pipeline for this task is shown in Fig. 4(b), where we use Lotus (He et al., 2024) to predict normal maps from the output of zero123++ (Shi et al., 2023a). Since the single-image-to-3D problem is inherently ill-posed, the unseen portions of the data cannot be accurately inferred from a single image alone. Consequently, we select 500 relatively clear data points for meaningful and valuable evaluation. In contrast, the reconstruction models, such as our DiMeR, PRM, and InstantMesh, have the advantages for the accurate alignment with input image based on the prediction of zero123++.

Table 2: Single-image-to-3D task. All the methods use the same single image input. Our DiMeR is equipped with Stable-Zero123++ (Shi et al., 2023a) and StableNormal (Ye et al., 2024).

| Dataset | GSO | | OmniObject3D | |
|---|---|---|---|---|
| Metric | CD (↓) | F1 (↑) | CD (↓) | F1 (↑) |
| CRM | 0.144 | 0.781 | 0.114 | 0.854 |
| MeshLRM | 0.079 | 0.933 | 0.086 | 0.915 |
| InstantMesh | 0.066 | 0.950 | 0.074 | 0.937 |
| PRM | 0.059 | 0.961 | 0.064 | 0.957 |
| DiMeR | 0.052 | 0.981 | 0.060 | 0.964 |

### 4.3 QUALITATIVE COMPARISON

**Sparse-view-to-3D.** As demonstrated in Fig. 5, we present a visual qualitative comparison of various methods. A comparison between the rows labeled "Ours" and "Ours (Lotus)" shows similar performance, highlighting that normal prediction models effectively support DiMeR. This suggests that DiMeR, when combined with such models, is capable of surpassing previous methods in real-

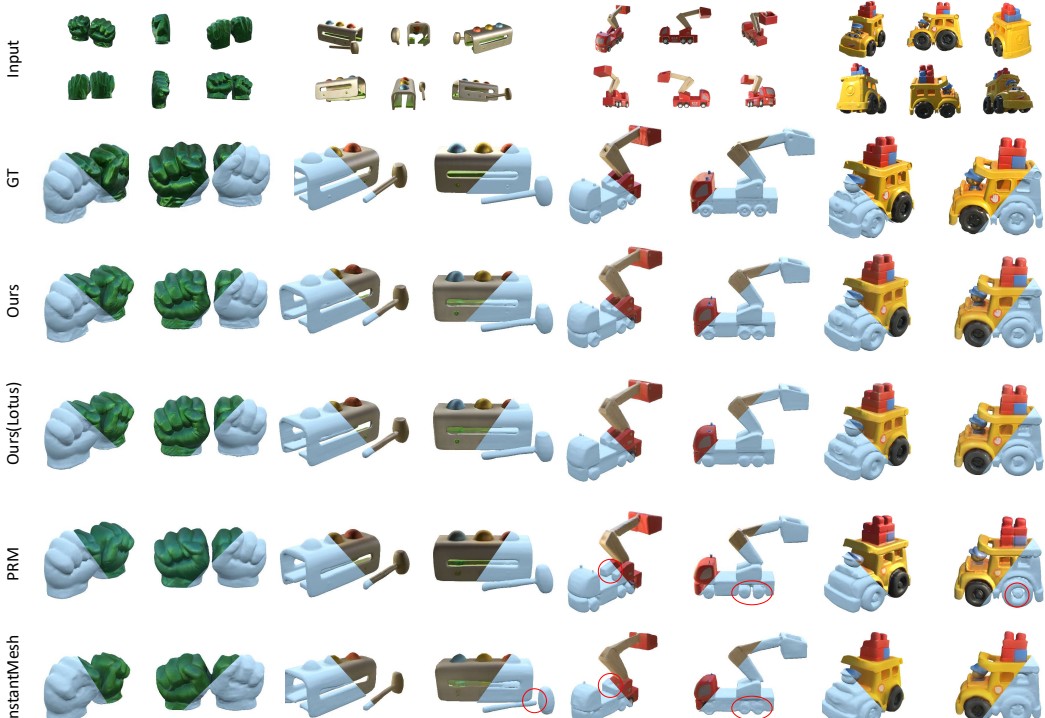

Figure 5: The qualitative comparison for sparse view reconstruction.

Table 3: The ablation studies of different input and output formats.

| Input & Output | CD($\downarrow$) | F1($\uparrow$) |
|---|---|---|
| RGB + Normal $\rightarrow$ Geometry + Texture | 0.043 | 0.976 |
| RGB + Normal $\rightarrow$ Geometry | 0.041 | 0.981 |
| Only Normal $\rightarrow$ Geometry (Ours) | 0.028 | 0.992 |

Table 4: The ablation studies of 3D regularization and PBR expectation.

| Input & Output | CD($\downarrow$) | F1($\uparrow$) |
|---|---|---|
| w/o 3D Regularization (Eq.1-2) | 0.037 | 0.975 |
| w/o PBR Expectation (Eq.3-4) | 0.039 | 0.973 |
| Our Full Design | 0.028 | 0.992 |

istic applications. Furthermore, DiMeR outperforms previous mesh reconstruction models, such as InstantMesh and PRM, in terms of reconstructing finer details.

**Single-image-to-3D.** As shown in Fig. 6, we compare our method with SoTA methods including Trellis (Xiang et al., 2024), PRM (Ge et al., 2024), MeshFormer (Liu et al., 2024), InstantMesh (Xu et al., 2024a) and CRM (Wang et al., 2025). Notably, since the 3D results for MeshFormer are only available from their project page and the corresponding input images are not provided, we are unable to conduct a direct comparison using the same input. In contrast, the other methods use the same input images for comparison. Due to the inherent nature of the generative in diffusion models, Trellis often generates 3D mesh models that exhibit inconsistencies with the input images, although it maintains high quality. Specifically, the cup's holes in the second column and the number of pillars in the third column are mismatched. Moreover, the other methods encounter difficulties in generating holes and rings accurately. In summary, DiMeR achieves the best quality.

### 4.4 ABLATION STUDIES

**Input & Output Disentanglement.** As shown in Tab. 13, we compare the different compositions of input and output for the geometry branch training. The comparison between the first two columns, "RGB + Normal $\rightarrow$ Geometry + Texture" and "RGB + Normal $\rightarrow$ Geometry", proves that the disentanglement for geometry and texture achieves a higher reconstruction accuracy for geometry. The comparison between the second and third columns, "RGB + Normal $\rightarrow$ Geometry" and "Only Normal $\rightarrow$ Geometry", demonstrates the huge gain for discarding the RGB input and supervision.

**Loss Design.** As demonstrated in Tab. 4, we show that Eq. 1 and Eq. 2 can replace the original loss functions used in FlexiCubes, performing improved performance. With the regularization loss employed in FlexiCubes, the training process becomes unstable and struggles to proceed beyond 10,000 iterations, resulting in unsatisfactory network convergence. By introducing the eikonal loss

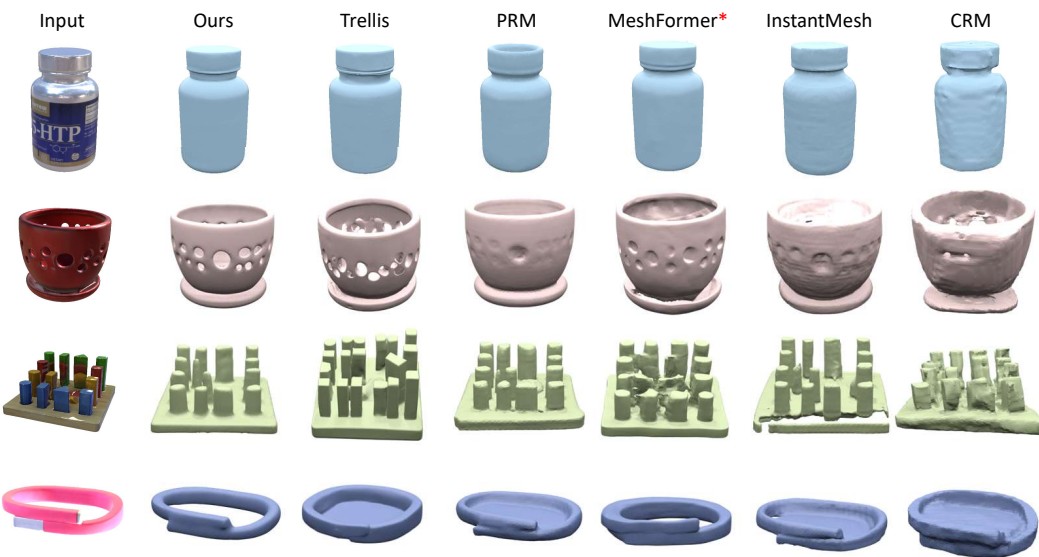

Figure 6: The qualitative comparison for single-image-to-3D. Please note that the results of Mesh-Former are obtained from their project page and do not use the same input as other methods.

and incorporating 3D ground truth, we stabilize the training process, achieving significantly better performance. We also validate the effectiveness of the PBR expectation loss (Eq. 3 and Eq. 4). If the lighting map can be accurately computed under varying environmental lighting conditions and across different materials, we can conclude that the predicted mesh aligns well with the ground truth mesh. To achieve this, we assign different materials to the single predicted mesh and place it in various environments. The introduction of PBR losses leads to significant improvements.

**Deformation and Weight MLP in FlexiCubes.** As shown in Tab. 5, we demonstrate that the improvements gained from the deformation and weight MLP are not worthy enough compared with their computational cost. The experiments are conducted using the official pre-trained weights of InstantMesh, and similar experiments based on PRM are provided in the supplementary material. Upon removing the deformation network and weight network from FlexiCubes, we observe minimal impact on inference performance, almost no decrease. However, these two networks significantly increase computational workload (about $2.5\times$ computation overhead) and GPU memory consumption (about $1.5\times$ GPU memory occupancy in training). Consequently, we opt to exclude them from DiMeR in order to improve the spatial resolution.

Table 5: The ablation studies of the effectiveness of Deformation and Weight MLP. GPU Mem is training occupancy.

| Method | CD | F1 | GPU Mem | Infer |
|--------|------|-------|---------|-------|
| w/ | 0.045 | 0.964 | 73GB | 0.5s |
| w/o | 0.045 | 0.963 | 48GB | 0.2s |

**Grid resolution.** As shown in Tab. 6, to validate the improvement, we add a comparison experiment at a 128 grid resolution. The experiment proves the improvement of our network architecture simplification. Furthermore, after reducing the resolution, our accuracy only decreased slightly and still remains higher than previous methods, InstantMesh and PRM.

## 4.5 MORE DISCUSSIONS.

**Reconstruction model vs Generative model.** As shown in Tab. 7 and Tab. 8, we compare trellis in three settings based on its original single-view version and official multi-view version. Even the multi-view version of the Trellis is still not a reconstruction model. Trellis is primarily designed as a generative model. Specifically, Trellis's multi-view input strategy involves randomly sampling viewpoints during different denoising steps. Only one randomly selected view is received at each denoising step. This cannot aggregate all views simultaneously, which may limit its ability to fully exploit multi-view consistency in our reconstruction setting. Moreover, Trellis can't generate 3D meshes that perfectly match the input images. These points explain its lower reconstruction accuracy under our evaluation protocol.

Table 6: Experiments about grid resolutions.

| Resolution | CD | F1 |
|---|---|---|
| 128 | 0.032 | 0.987 |
| 192 | 0.028 | 0.992 |

Table 7: Comparison with Trellis on multi-view.

| Method | CD(↓) | F1(↑) |
|---|---|---|
| Trellis(MV) | 0.136 | 0.835 |
| DiMeR(GT) | 0.028 | 0.992 |
| DiMeR(SN) | 0.032 | 0.988 |

Table 8: Comparison with Trellis on single-view.

| Method | CD(↓) | F1(↑) |
|---|---|---|
| Trellis original | 0.119 | 0.859 |
| Trellis+zero123plus | 0.155 | 0.783 |
| DiMeR | 0.052 | 0.981 |

Table 9: Quantitative results for reconstruction task with different normal predictors. CD means Chamfer Distance. Error is the angle between predicted normal vectors and gt normal vectors.

| Dataset | GSO | | | OmniObject3D | | |
|---|---|---|---|---|---|---|
| Metric | CD (↓) | F1 (↑) | Error (↓) | CD (↓) | F1 (↑) | Error (↓) |
| GeoWizard+DiMeR | 0.039 | 0.982 | 17.673 | 0.037 | 0.987 | 23.129 |
| MariGold+DiMeR | 0.037 | 0.986 | 17.400 | 0.038 | 0.984 | 22.934 |
| DSINE+DiMeR | 0.040 | 0.978 | 17.953 | 0.046 | 0.974 | 23.010 |
| Lotus-G+DiMeR | 0.033 | 0.988 | 17.151 | 0.036 | 0.988 | 21.523 |
| Lotus-D+DiMeR | 0.035 | 0.987 | 16.606 | 0.034 | 0.989 | 21.065 |
| StableNormal+DiMeR | 0.032 | 0.988 | 16.818 | 0.030 | 0.993 | 21.205 |

Table 10: Different normal sources for single-image-to-3D task with Zero123plus.

| Method | CD(↓) | F1(↑) |
|---|---|---|
| DiMeR + Zero123plus joint Normal | 0.061 | 0.960 |
| DiMeR + Zero123plus SN Normal | 0.052 | 0.981 |

Table 11: Robustness to Gaussian noise in normal maps.

| Noise std | CD(↓) | F1(↑) |
|---|---|---|
| 0.0 | 0.028 | 0.992 |
| 0.05 | 0.031 | 0.991 |
| 0.1 | 0.032 | 0.991 |
| 0.3 | 0.036 | 0.987 |
| 0.5 | 0.042 | 0.982 |

**Robustness on different normal prediction models.** As shown in Tab. 9, we conduct the experiments based on different normal foundation models as DiMeR's input. DiMeR empirically shows strong robustness across different normal prediction models. Furthermore, we can observe that the CD value exhibits a decreasing trend as the normal prediction error decreases.

Besides different normal prediction models, we also conduct experiments on the joint normal maps output from zero123plus Shi et al. (2023a). As shown in Tab 10, these results indicate the robustness of our DiMeR can adapt to any normal source.

Finally, as shown in Fig. 11, we conduct the experiments on giving some noise to the objects' normal maps.

## 5 CONCLUSION

In this paper, we propose DiMeR, a disentangled dual-stream framework with 3D supervision for feed-forward sparse-view mesh reconstruction. By driving the geometry branch exclusively with normal maps and leaving RGB information to a separate texture branch, DiMeR clearly separates conflicting objectives and grounds training on unambiguous supervision signals. To enhance the training effectiveness and spatial resolution, DiMeR improves the mesh extraction algorithm by redesigning the regularization losses, introducing 3D ground-truth supervision, and removing redundant modules. Extensive experiments confirm that DiMeR surpasses state-of-the-art baselines across multiple tasks, such as sparse-view-to-3D, image-to-3D, and text-to-3D, highlighting both its effectiveness and robustness. As normal-prediction models continue to improve, DiMeR's performance is likely to advance further.

**Limitations.** 1) Since we train DiMeR on the object-wise normal maps with a white background, we can not accept the scene-level sparse views as input. 2) On the single-image-to-3D task, the enhancement of the 2.5D diffusion model, including resolution and accuracy, would help DiMeR achieve better quality.

## ACKNOWLEDGMENTS

This project is partially supported by National Key R&D Program of China (Grant No.2023YFF0725001).

## REPRODUCIBILITY STATEMENT

We have uploaded the complete project code in the supplementary materials to ensure the reproducibility of our work. Additionally, the results presented for all other methods were obtained directly from their official repositories, ensuring that they are accurately reflected as per their published implementations.

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

## A    IMPLEMENTATION DETAILS

**Evaluation Protocol.** For 3D metrics, we sample $32,000$ points on the surface to compute commonly used Chamfer Distance (CD) and F1-Score@0.1 to evaluate the quality of geometry. For 2D metrics, we compute the PSNR, SSIM, and LPIPS to evaluate the rendering quality over $8$ rendered views. We rescale and align the generated meshes and ground truth meshes for fair comparison.

**Training.** We set the total batch size to $64$, with learning rate of $4 \times 10^{-6}$ for geometry branch and $4 \times 10^{-5}$ for texture branch. The resolution of the triplane is $3 \times 64 \times 64$, and the SDF grid is $192^3$, which is higher than baselines benefiting from our enhancement of mesh extraction methods. The resolutions of input and supervision are $512 \times 512$. For PBR statistical expectation loss, we place the predicted meshes in 10 different lighting environments and apply 10 different materials, rendering from different 10 views during training. We train the geometry branch for two days and the texture branch for one day on 8 H100 GPUs.

## B    BENCHMARK FOR NORMAL PREDICTION MODELS

To determine whether recent normal-prediction foundation models meet the quality requirements of our pipeline, we evaluate Lotus (He et al., 2024), StableNormal (Ye et al., 2024), DSINE (Bae & Davison, 2024), Marigold (Ke et al., 2024), and GeoWizard (Fu et al., 2024) on the GSO (Downs et al., 2022) and OmniObject3D (Wu et al., 2023) benchmarks. For each object, six randomly sampled views are rendered, producing paired RGB images, masks, and ground-truth normal maps. Using the RGB inputs, we measure mean and median angular error, the proportion of pixels with error below $11.25°$, $22.5°$, and $30°$, and inference time. When computing these metrics, we only use the foreground pixels. As summarised in Tab. 12, StableNormal and Lotus offer the best balance of accuracy and speed, adding only negligible latency. Correspondingly, as demonstrated in Tab. 1, even with errors, **our DiMeR still outperforms previous methods by a large margin, accepting the same RGB input**. Among the reported metrics, the mean error and the fraction of pixels with error below the threshold $30°$ are most indicative of prediction stability. Large errors markedly impact reconstruction quality. Ongoing advances in normal-prediction models are therefore expected to further improve DiMeR's performance. We also provide the qualitative comparison in Fig. 7.

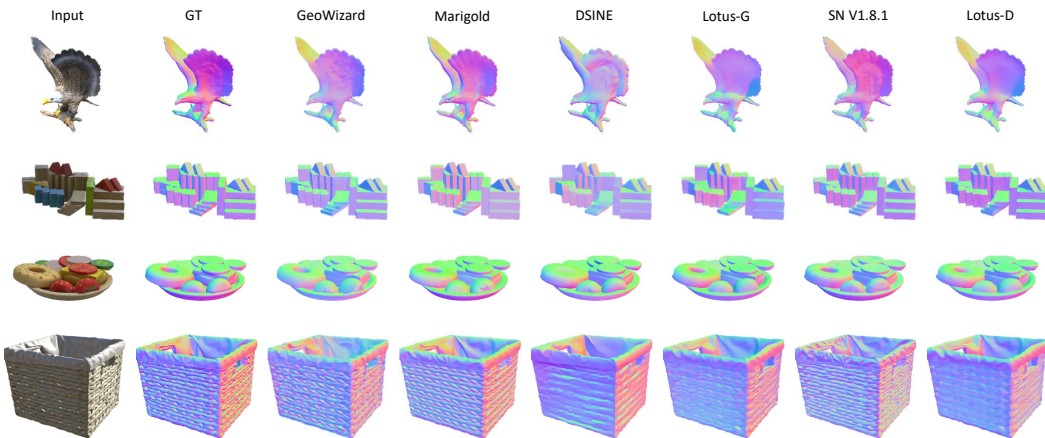

Figure 7: The qualitative comparison for normal prediction foundation models.

Table 12: Benchmark for normal map prediction of foundation models on object scenario. The latency is evaluated on a single A800 GPU.

| Dataset | GSO | | | | | OmniObject3D | | | | | - |
|---|---|---|---|---|---|---|---|---|---|---|---|
| Metric | mean (↓) | median (↓) | 11.25° (↑) | 22.5° (↑) | 30° (↑) | mean (↓) | median (↓) | 11.25° (↑) | 22.5° (↑) | 30° (↑) | Latency (↓) |
| GeoWizard | 17.673 | 14.307 | 48.309 | 74.908 | 83.097 | 23.129 | 20.156 | 28.272 | 61.215 | 74.122 | 2102 ms |
| Marigold | 17.400 | 14.305 | 47.303 | 76.058 | 84.197 | 22.934 | 20.243 | 28.867 | 61.201 | 73.824 | 260 ms |
| DSINE | 17.953 | 14.857 | 45.543 | 72.951 | 82.535 | 23.010 | 20.116 | 29.182 | 62.140 | 75.082 | 59 ms |
| Lotus-G | 17.151 | 13.920 | 45.343 | 76.831 | 85.277 | 21.523 | 19.048 | 30.836 | 64.828 | 77.359 | 130 ms |
| SN V1.8.1 | 16.818 | 14.743 | 39.860 | 74.524 | 86.424 | 21.205 | 19.468 | 25.677 | 61.917 | 77.916 | 236 ms |
| Lotus-D | 16.606 | 13.377 | 47.218 | 78.076 | 86.166 | 21.065 | 18.622 | 32.118 | 66.216 | 77.968 | 130 ms |

## C DISCUSSION ABOUT THE IMPORTANCE OF NORMAL-ONLY

A normal map is simply an image whose three channels no longer describe colour but the $x,y$ and $z$ components of the unit normal vector at every pixel. In practice each component, which lies in $[-1, 1]$, is linearly re-encoded into the 0-255 range and written into the R-,G- and B-channels. When this map is projected back onto the mesh under the same camera pose, every RGB triplet can be decoded to recover the outward-facing surface normal at that point. Because it is a dense field of orientations, the map tells us "how the surface tilts here" rather than "what colour it is here". In other words, it provides first-derivative information about shape—much like having a slope-at-every-point description of a landscape.

Specifically, training on RGB images introduces substantial ambiguity. The network must infer whether changes in colour are due to actual shape (e.g., surface relief) or merely superficial appearance (e.g., paint or lighting). As illustrated in Figure 2 of the paper, visually identical features—such as black dots on a die—may stem from entirely different causes: painted texture or recessed geometry. This ambiguity forces the network to learn additional heuristics, increasing inductive bias and complicating generalisation. Normal maps eliminate this burden. Because they encode only surface orientation, no appearance disentanglement is needed. The network can instead learn a more direct and well-conditioned mapping from normals to 3D shape. Empirically, this reduced-bias setting yields clear benefits: substituting RGB supervision with normal maps improves Chamfer distance from 0.041 to 0.028 and raises the F1 score from 0.971 to 0.992. These results confirm that normal-based learning is not only simpler, but also more effective.

Table 13: The ablation studies of different input formats.

| Input & Output | CD($\downarrow$) | F1($\uparrow$) |
|---|---|---|
| RGB | 0.041 | 0.971 |
| RGB + Normal | 0.041 | 0.981 |
| Normal | 0.028 | 0.992 |

## D ABLATION STUDIES FOR DEFORMATION AND WEIGHT NETOWRKS

To validate the inefficiency of deformation and weight networks in FlexiCubes, we additionally conduct the experiments based on the official pre-trained weights of PRM. As shown in Tab. 14, removing these two networks doesn't damage the performance. Therefore, the experiments of Tab. 5 and Tab. 14 can prove our conclusion, the deformation network and weight network are redundant for LRM series models. Cutting the cost of these two modules, we can improve the spatial resolution from $128^3$ to $192^3$. Moreover, the latency of each iteration in training is also reduced.

Table 14: The ablation studies of the effectiveness of Deformation and Weight MLP based on PRM. GPU Mem is training occupancy.

| Method | CD | F1 |
|---|---|---|
| w/ | 0.041 | 0.977 |
| w/o | 0.041 | 0.977 |

## E DISCUSSION ABOUT THE INFLUENCE OF NORMAL PREDICTION ACCURACY

In DiMeR's training, we take the unstable prediction from normal prediction models into consideration. Specifically, we add noise to the GT normal input to improve robustness in practical utilization. Though the slight accuracy drop is inevitable, the overall accuracy is still much higher than the baseline's. As shown in Tab. 1, when replacing the GT normal map from the output of Lotus or StableNormal, the geometry accuracy only reduces by 0.004 (from 0.028 to 0.032), which gains 22.0% compared with the previous SoTA model, PRM.

## F DISCUSSION ABOUT LOSS DESIGN

All of Eq.3-7 are texture-irrelevant. Normal, Depth, Masks, and PBR light map losses are 2.5D, which can provide a higher spatial resolution supervision. Since the unaffordable computational burden when increasing 3D spatial resolution, the 3D SDF loss mainly fuction on a regularization of

the space ($192^3$), such as the object's internal. Therefore, the combination of high-resolution 2.5D supervision with relatively low-resolution 3D regularization is more reasonable.

## G MORE VISUAL RESULTS FOR SPARSE VIEW TO 3D

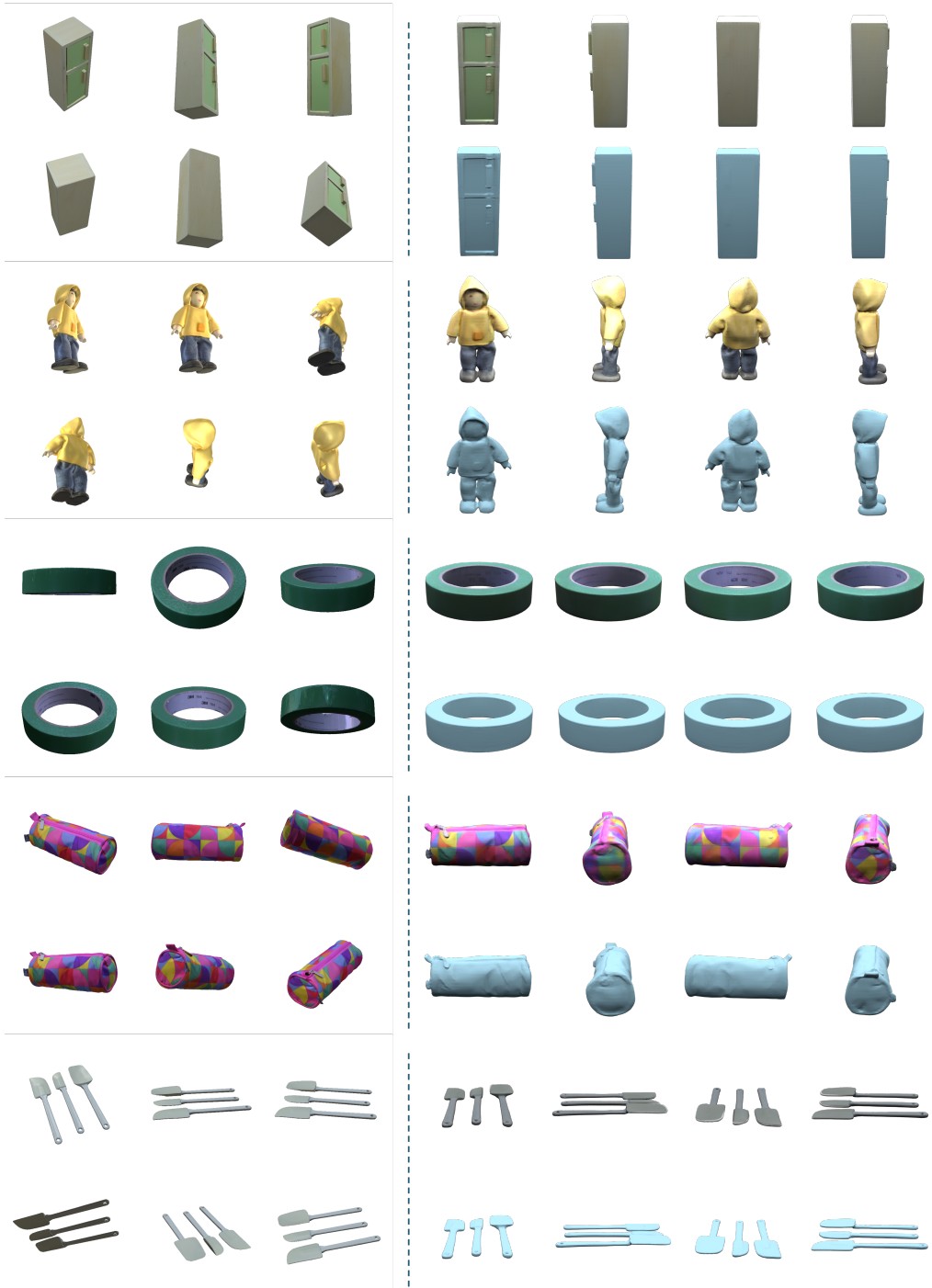

Figure 8: More Visual Results for Sparse View to 3D. The left and right are input and output seperately.

## H   MORE VISUAL RESULTS FOR TEXT TO 3D

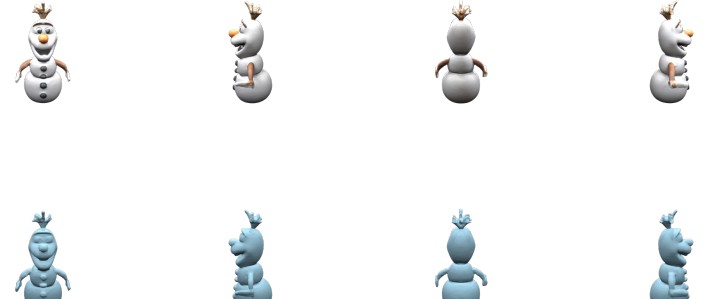

"Olaf the Snowman three stacked spheres, proportionally larger base sphere, medium mid-section sphere, smaller head sphere, protruding conical nose, two dot-like eyes, three buttons aligned vertically, twig-like hair feature, two stick-like arms, curvilinear shapes"

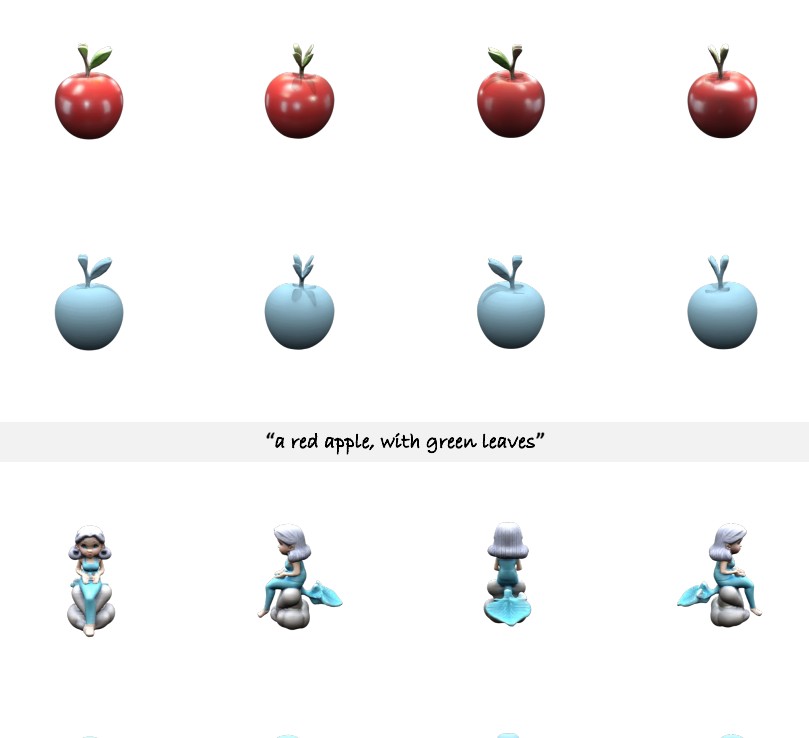

"a red apple, with green leaves"

"A mermaid with a blue dress and a koi fish tail sitting on a stone"

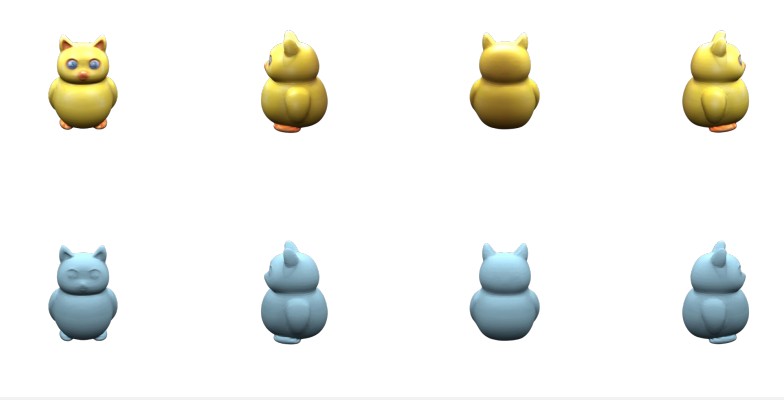

"A yellow owl and cat-shaped toy"

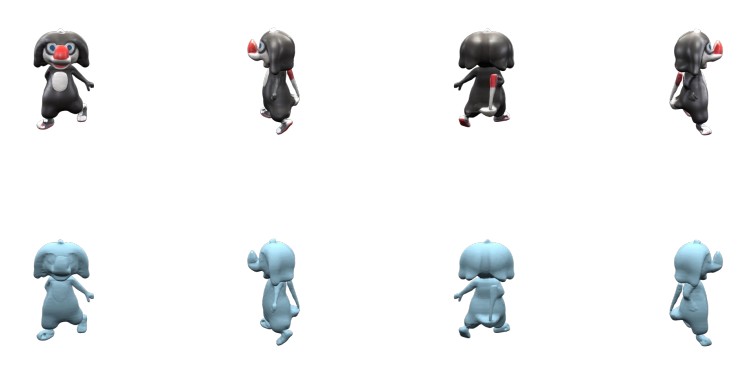

"a black and white cartoon character with a red mouth and nose"

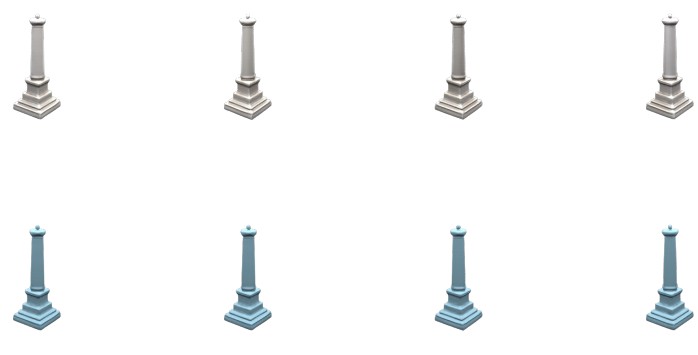

"a stone monument on top of a pedestal"

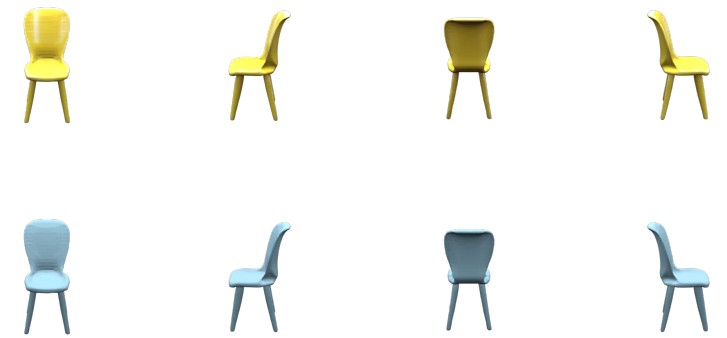

"a yellow plastic chair with a curved backrest"

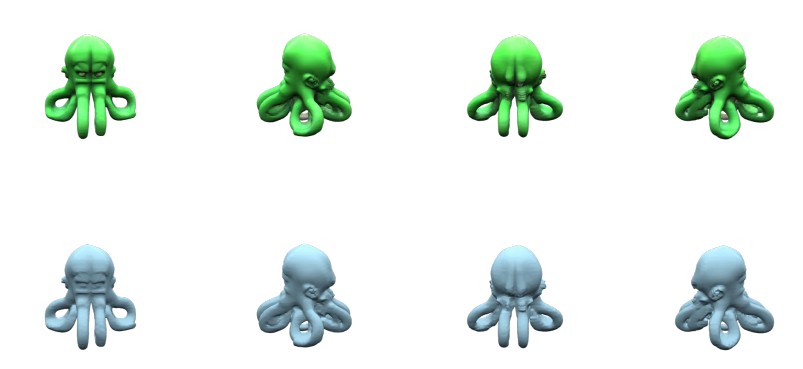

"a green octopus with tentacles and a Cthulhu head"

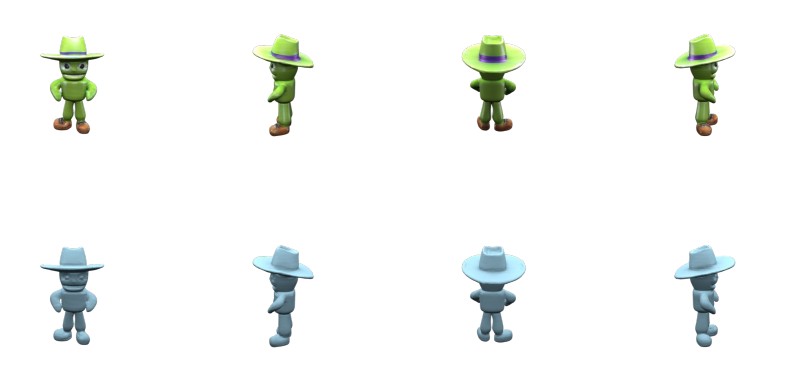

"a humanoid Mexican cactus wearing a green and purple hat"

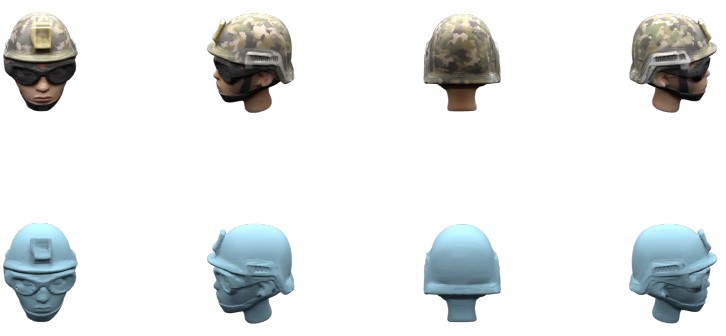

"A head wearing a helmet and goggles"

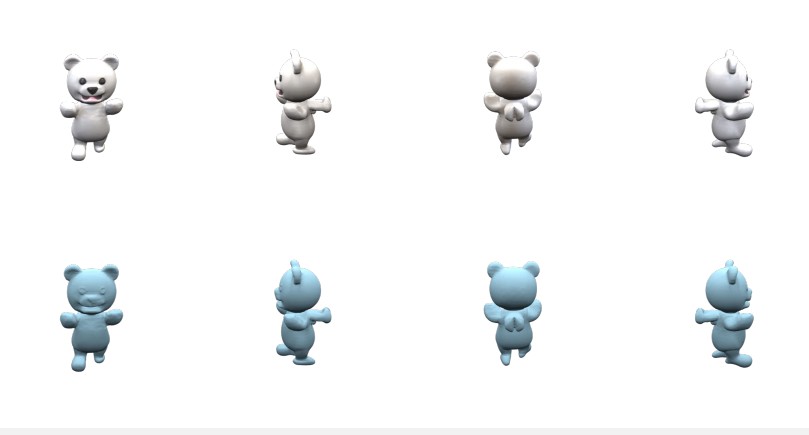

"a white teddy bear"

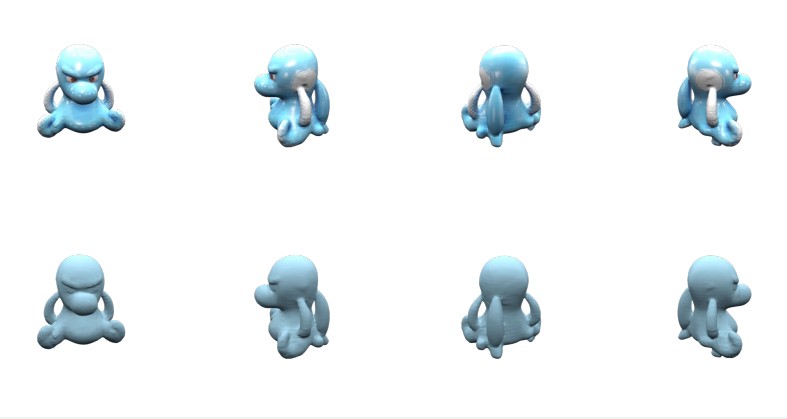

"a blue and white octopus-like creature"

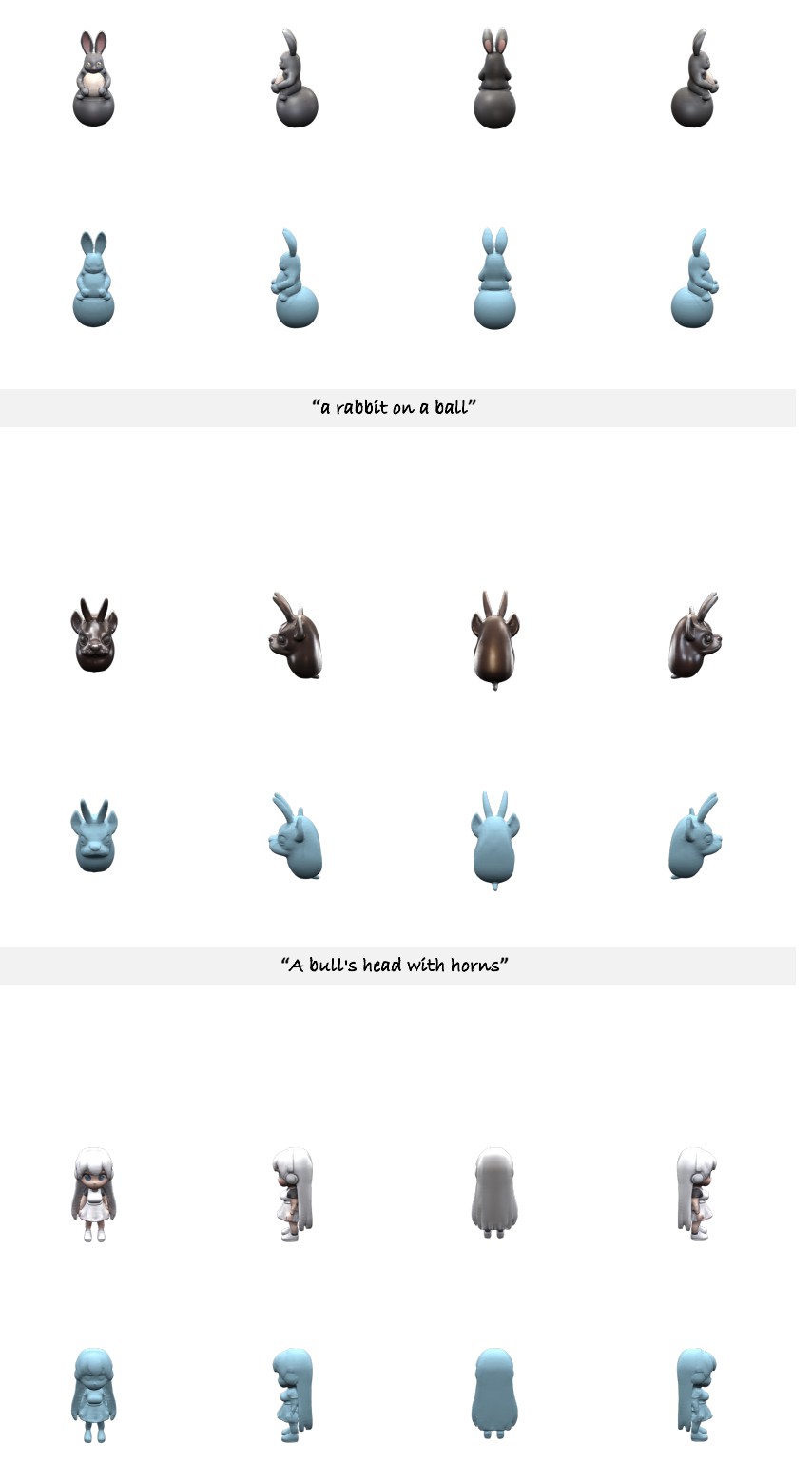

"a rabbit on a ball"

"A bull's head with horns"

"a girl with long white hair, wearing a white dress and headphones"

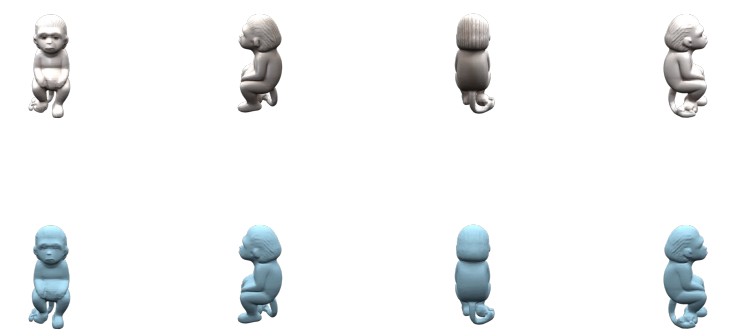

"A small white stone sculpture of a monkey, seated posture, delineated mane"

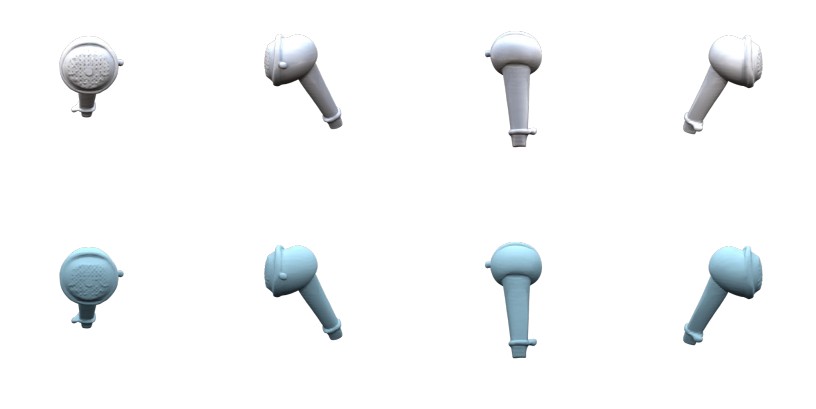

"White shower head"

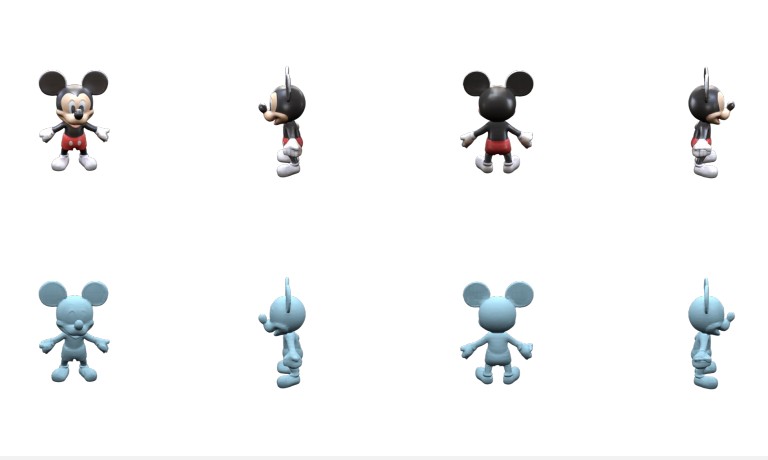

"A cartoon Mickey Mouse"

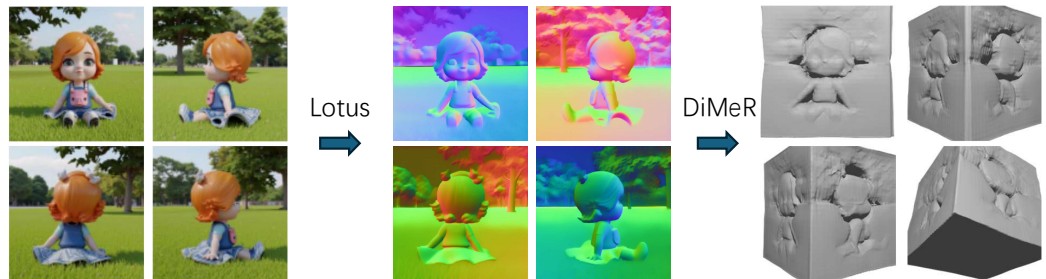

Figure 9: A failure case.

## I FAILURE CASE

Since we train DiMeR on the object-level normal maps with a white background, we can not accept the image with a scene background as input. As shown in Fig. 9, DiMeR fail to synthesis the mesh model for a scene-level input.

## J PBR SETUP

We use a microfacet BRDF with a Lambertian diffuse term and a Cook–Torrance specular term, with a Trowbridge–Reitz GGX normal distribution, Schlick Fresnel, and a GGX-based geometry term. Materials are parameterized in the usual metalness–roughness form: per-pixel albedo, metallic, and roughness. For lighting, we rely on image-based lighting with HDR environment maps. At training time, each rendered view uses a randomly sampled environment map from a large collection of 679 indoor and outdoor HDRIs. We use the standard split-sum approximation: the diffuse term is computed from a low-frequency irradiance map, while the specular term is obtained from a prefiltered mipmapped environment map. The range of roughness and metallic parameters is [0,1].

## K THE USE OF LARGE LANGUAGE MODELS (LLM)

We used OpenAI's GPT-5 to assist with the refinement and proofreading of certain sentences in this paper. The LLM was used exclusively to enhance the clarity and coherence of our writing. All content contributions are made by the authors.

