# OpenReview forum: "DiMeR: Disentangled Mesh Reconstruction Model with Normal-only Geometry Training"
_ICLR.cc/2026/Conference — ICLR 2026 Poster_

### Official Review · Reviewer_SAKn · 2025-10-29

**Soundness:** 3
**Presentation:** 3
**Contribution:** 3
**Rating:** 6
**Confidence:** 4

**Summary:**

This paper proposes DiMeR, a 3D mesh reconstruction framework that disentangles geometry and texture to reduce training ambiguity. It predicts geometry solely from normal maps while learning texture from RGB images in a separate branch. The authors further improve the pipeline with techniques like Eikonal and PBR expectation losses, and demonstrate significant performance gains across sparse-view, single-image, and text-to-3D tasks. The overall design is clean, stable, and achieves clear improvements over prior methods.

**Strengths:**

- The paper introduces a novel framework that reconstructs geometry purely from normal maps, effectively disentangling geometry and texture spaces, (partially) resolving the ambiguity between shape and appearance in RGB-supervised 3D reconstruction.
- The method is well-engineered, integrating improvements like Eikonal regularization and a PBR expectation loss. The latter is especially novel as it introduces varying lighting and material as a way of disentangling geometry and appearance.
- The authors conducted extensive experiments on GSO and OmniObject3D, showing substantial improvements over prior methods. Comprehensive ablation studies confirm the effectiveness of each design choice, demonstrating strong technical rigor.

**Weaknesses:**

- In the Single-Image-to-3D and Text-to-3D experiments, the authors only use StableNormal and Lotus to generate normals, while models like Zero123++ can already output RGB and normal maps jointly. To validate robustness, the paper should test multiple normal sources (e.g., Zero123++, Era3D, Kiss3DGen) and report Chamfer/F1 variations under different normal qualities.
- The model is trained with ground-truth normals but tested with predicted normals, which may lead to a training-testing gap. The authors are encouraged to train with noisy or predicted normals to better reflect real-world robustness.
- Missing noise sensitivity analysis. The paper should include an additional experiment that perturbs GT normals with controlled noise levels and compares quantitative and qualitative results, to show how geometry accuracy degrades under realistic prediction errors.

**Questions:**

- The specular rendering loss in the proposed PBR expectation term may suffer from gradient instability when the roughness parameter is small, as specular reflections can become highly discontinuous. Could the authors clarify how they stabilize this loss or whether any regularization is applied?
- It would also be helpful to provide more details about the PBR setup — specifically, the BRDF model used, the range of roughness and metallic parameters, and the nature of the lighting conditions (e.g., sharp high-frequency environment maps vs. smooth diffuse ones).
- Additionally, from the released code, it seems that only a single environment map is used for rendering; could the authors confirm whether multiple environments were actually sampled during training, as stated in the paper?

---

> ### Author Response · Authors · 2025-11-18
> **Response to Reviewer SAKn (1/2)**
>
> We **sincerely** thank you for reviewing our paper and for your constructive comments. We have carefully addressed all the weaknesses and questions raised, providing additional experiments and detailed explanations. **The revisions have been highlighted in cyan in the updated manuscript.** We hope our responses adequately address your concerns, and we remain available for any further discussion or clarification you may require.
>
> ### **[W1] Results on more normal prediction models**
>
> Thank you for your suggestion. To test multiple normal sources, we conduct the experiments using different normal sources.
>
> **Firstly**, we incorporate more experiments using more normal prediction models and **update it to Sec. 4.5 of the revised paper (noted in cyan color)**.
>
> On the GSO dataset, the results are
>
> |                    | Chamfer Distance(↓) | F1-Score(↑) | Normal Prediction Mean Error(↓) |
> |--------------------|:-------------------:|:-----------:|:-------------------------------:|
> | GeoWizard+DiMeR    |        0.039        |    0.982    |              17.673             |
> | MariGold+DiMeR     |        0.037        |    0.986    |              17.400             |
> | DSINE+DiMeR        |        0.040        |    0.978    |              17.953             |
> | Lotus-G+DiMeR      |        0.033        |    0.988    |              17.151             |
> | Lotus-D+DiMeR      |        0.035        |    0.987    |              16.606             |
> | StableNormal+DiMeR |        0.032        |    0.988    |              16.818             |
>
>
> On the OmniObject3D, the results are
>
> |                    | Chamfer Distance(↓) | F1-Score(↑) | Normal Prediction Mean Error(↓) |
> |--------------------|:-------------------:|:-----------:|:-------------------------------:|
> | GeoWizard+DiMeR    |        0.037        |    0.987    |              23.129             |
> | MariGold+DiMeR     |        0.038        |    0.984    |              22.934             |
> | DSINE+DiMeR        |        0.046        |    0.974    |              23.010             |
> | Lotus-G+DiMeR      |        0.036        |    0.988    |              21.523             |
> | Lotus-D+DiMeR      |        0.034        |    0.989    |              21.065             |
> | StableNormal+DiMeR |        0.030        |    0.993    |              21.205             |
>
>
> **Secondly**, on the single-image-to-3D task, we conduct the experiments based on the joint normal from zero123plus and **update it to Sec. 4.5 of the revised paper (noted in cyan color)**. These results indicate the robustness of our DiMeR can adapt to any normal source.
>
> |                                  | Chamfer Distance(↓) | F1-Score(↑) |
> |----------------------------------|:-------------------:|:-----------:|
> | DiMeR + Zero123plus joint Normal |        0.061        |    0.960    |
> | DiMeR + Zero123plus SN Normal    |        0.052        |    0.981    |
>
>
> **Thirdly**, we provide extensive text-to-3D results based on Kiss3DGen in the appendix.
>
> ### **[W2] Training strategy**
>
> Thank you for your suggestion. During training, our model already adds slight noise (part iterations) to the input normal maps. Since different normal prediction models each have their own biases, and we want to support all of them, we train using only GT normals with added slight noise. This strategy is justified because: (1) it allows our model to learn the underlying geometric structures without being biased toward specific prediction errors; (2) the added noise during training helps bridge the training-testing gap; (3) our extensive experiments show that this approach generalizes well to various prediction models with different error characteristics.

---

> ### Author Response · Authors · 2025-11-18
> **Response to Reviewer SAKn (2/2)**
>
> ### **[W3] Robustness on noisy input**
>
> Thanks for your suggestions. We additionally conducted experiments to evaluate the model by adding different levels of Gaussian noise to the input normal maps. **Moreover, we update it to Sec. 4.5 of the revised paper (noted in cyan color).**
>
> | std  | Chamfer Distance(↓) | F1-Score(↑) |
> |------|:-------------------:|:-----------:|
> | 0.0  |        0.028        |    0.992    |
> | 0.05 |        0.031        |    0.991    |
> | 0.1  |        0.032        |    0.991    |
> | 0.3  |        0.036        |    0.987    |
> | 0.5  |        0.042        |    0.982    |
>
> ### **[Q1] Stability of specular rendering loss**
>
> We perform not only pixel-level loss calculations but also feature-level loss computations. In such extreme cases, the LPIPS loss can provide global stability.
>
> ### **[Q2] More details about PBR setup**
>
> Thank you for your suggestions and **we have update this in the last of the revised paper (noted in cyan color).** Our PBR setup is as follows:
>
> We use a microfacet BRDF with a Lambertian diffuse term and a Cook–Torrance specular term, with a Trowbridge–Reitz GGX normal distribution, Schlick Fresnel, and a GGX-based geometry term. Materials are parameterized in the usual metalness–roughness form: per-pixel albedo, metallic, and roughness. For lighting, we rely on image-based lighting with HDR environment maps. At training time, each rendered view uses a randomly sampled environment map from a large collection of 679 indoor and outdoor HDRIs. We use the standard split-sum approximation: the diffuse term is computed from a low-frequency irradiance map, while the specular term is obtained from a prefiltered mipmapped environment map. The range of roughness and metallic parameters is [0,1].
>
> ### **[Q3] About the environment map**
>
> Thank you for pointing this out. We apologize for any confusion caused by the released code. The current release includes only the inference code with a placeholder environment map for simplicity. During training, we randomly sample from a collection of 679 HDR environment maps, enabling diverse lighting conditions. The full training code with the complete environment map collection will be released upon paper acceptance.

---

> ### Author Response · Authors · 2025-11-27
>
> Dear Reviewer SAKn,
>
> Thank you once again for your valuable feedback. We have conducted additional experiments and made revisions to the paper based on your suggestions. **As the discussion phase is nearing its conclusion**, we would like to know if our responses have addressed your concerns. We are looking forward to hearing from you.
>
> Best, Authors

---

### Official Review · Reviewer_jDrM · 2025-10-30

**Soundness:** 3
**Presentation:** 3
**Contribution:** 3
**Rating:** 6
**Confidence:** 3

**Summary:**

The paper introduces a disentangled dual-branch framework (DiMeR) for 3D mesh reconstruction. Unlike previous approaches that jointly train the texture and geometry space, DiMeR separates the geometry and texture learning using only normal maps and RGB inputs. It proposes geometry-specific losses and overall achieves large performance gains on two benchmarks for sparse-view, single-view and text-to-3D reconstruction.

**Strengths:**

1. The motivation is quite clear and interestingly stated in Figure 2. The disentanglement is clean and effective. The performance improvement in Table 3 demonstrates the effectiveness of the proposed approach.
2. The pruning of heavyweight FlexiCubes components achieves faster training and fewer overhead on GPU memory (in Table 5), which allows higher resolution without accuracy loss
3. The evaluation is conducted comprehensively, including different modules in the ablations and comparison with previous baselines on 3 applications over two benchmarks.

**Weaknesses:**

The architecture is a bit like the combination of LRM with FlexiCubes. The backbone largely follows existing LRM-style architecture. Most novelty lies in the training strategy and disentanglement of texture and geometry. But overall, the adaptation with obvious improvement, especially on geometry, is promising.

**Questions:**

1. Is there any approach to measure how sensitive DiMeR's geometry performance is to errors in normal prediction?
2. Since the geometry and texture branches are disentangled, how is consistency ensured during joint inference? Is it totally influenced by the normal map consistency with the RGB input, or any corner cases when the predictions of these two branches are not aligned well?

---

> ### Author Response · Authors · 2025-11-18
> **Response to Reviewer jDrM**
>
> We **sincerely** thank you for reviewing our paper and for your constructive comments. We have carefully addressed all the weaknesses and questions raised, providing additional experiments and detailed explanations. **The revisions have been highlighted in cyan in the updated manuscript.** We hope our responses adequately address your concerns, and we remain available for any further discussion or clarification you may require.
>
> ### **[W] Novelty**
>
> We appreciate your recognition of the promising improvements, especially in geometry reconstruction. While our architecture builds upon proven components like LRM and FlexiCubes, we believe the key contribution lies in **a stronger inductive bias via strict modality-level *input* disentanglement for the elimination of ambiguity and hallucination in training (shown in Fig.2).**
>
> ### **[Q1] Sensitivity to normal prediction**
>
> Thank you for your suggestion. Following your advice, we add an evaluation of how incorporating different normal prediction models affects the accuracy of our geometric reconstruction and and **update it to Sec. 4.5 of the revised paper (noted in cyan color)**. Since these normal foundation models exhibit errors to varying degrees, and the types of errors also differ across models — some being overly smooth and others overly sharp — this can be regarded as mimicking different types of errors.
>
> On the GSO dataset, the results are
>
> |                    | Chamfer Distance(↓) | F1-Score(↑) | Normal Prediction Mean Error(↓) |
> |--------------------|:-------------------:|:-----------:|:-------------------------------:|
> | GeoWizard+DiMeR    |        0.039        |    0.982    |              17.673             |
> | MariGold+DiMeR     |        0.037        |    0.986    |              17.400             |
> | DSINE+DiMeR        |        0.040        |    0.978    |              17.953             |
> | Lotus-G+DiMeR      |        0.033        |    0.988    |              17.151             |
> | Lotus-D+DiMeR      |        0.035        |    0.987    |              16.606             |
> | StableNormal+DiMeR |        0.032        |    0.988    |              16.818             |
>
>
> On the OmniObject3D, the results are
>
> |                    | Chamfer Distance(↓) | F1-Score(↑) | Normal Prediction Mean Error(↓) |
> |--------------------|:-------------------:|:-----------:|:-------------------------------:|
> | GeoWizard+DiMeR    |        0.037        |    0.987    |              23.129             |
> | MariGold+DiMeR     |        0.038        |    0.984    |              22.934             |
> | DSINE+DiMeR        |        0.046        |    0.974    |              23.010             |
> | Lotus-G+DiMeR      |        0.036        |    0.988    |              21.523             |
> | Lotus-D+DiMeR      |        0.034        |    0.989    |              21.065             |
> | StableNormal+DiMeR |        0.030        |    0.993    |              21.205             |
>
> ### **[Q2] How to ensure consistency**
>
> Since we are not directly predicting the color of each vertex, but instead predicting a texture field, the texture field provides a certain tolerance to errors: even if the vertex coordinates are slightly shifted, the interpolated features sampled from the texture field will not change much. Consequently, the resulting colors will also remain largely stable. Therefore, the impact of vertex coordinate errors on the texture field is relatively limited.
>
> In addition, we train this branch together with the output of the geometry branch. As a result, the texture branch learns to adapt to the errors in the geometry branch.

---

> ### Author Response · Authors · 2025-11-27
>
> Dear Reviewer jDrM,
>
> Thank you once again for your valuable feedback. We have conducted additional experiments and made revisions to the paper based on your suggestions. **As the discussion phase is nearing its conclusion**, we would like to know if our responses have addressed your concerns. We are looking forward to hearing from you.
>
> Best, Authors

---

### Official Review · Reviewer_QJa6 · 2025-11-02

**Soundness:** 3
**Presentation:** 3
**Contribution:** 2
**Rating:** 4
**Confidence:** 4

**Summary:**

The paper proposes DiMeR, a feed-forward mesh reconstruction framework that explicitly disentangles geometry and texture: the geometry branch takes only surface normal maps; the texture branch takes RGB and is supervised by appearance losses. The geometry branch still accepts raw RGB at inference by first predicting normals with recent "foundation" normal estimators (e.g., Lotus, StableNormal). The authors also simplify FlexiCubes by removing deformation/weight MLPs and swap its regularizers for eikonal + GT-SDF supervision, allowing a higher resolution SDF grid at similar cost. Experiments on GSO and OmniObject3D report better geometry quality versus InstantMesh and PRM for sparse-view, single-image, and text-to-3D pipelines.

**Strengths:**

1. The paper effectively illustrates the texture-hides-geometry problem in Figure 2, motivates normal-only geometry prediction to avoid multi-solution conflicts; the design is well illustrated and methodically grounded.
2. Clear technical contributions e.g. replacing FlexiCubes regularizers with eikonal + GT-SDF supervision and adding PBR expectation losses. These modifications are technically sensible and addresses known stability issues.
3. Empirical gains on GSO / OmniObject3D. DiMeR shows good amount of CD improvements. Single-image comparisons also improve over InstantMesh / PRM. Results remain strong even with predicted normals.
4. Comprehensive evaluation. Tests across multiple tasks (sparse-view, single-image, text-to-3D). Ablations isolate the benefit of normal-only input and 3D regularization + PBR losses; FlexiCubes pruning demonstrates little accuracy loss but notable memory/latency savings. These provide valuable insights.

**Weaknesses:**

1. Novelty relative to contemporaries is moderate. Prior work has: 1) disentangled geometry/texture for 3D generation (e.g., Fantasia3D/CLAY), 2) used normals as strong geometry cues (PRM, photometric-stereo based LRM), and 3) pursued feed-forward reconstruction with triplanes/meshes (LRM family, InstantMesh, MeshLRM, MeshFormer). The paper should delineate what is fundamentally new beyond combining these threads, clearer positioning vs MeshLRM/MeshFormer would help.
2. Fairness/controls in single-image pipeline. For single-image-to-3D, the pipeline uses Zero123++ -> Lotus normals -> DiMeR and claims all reconstruction baselines use the same Zero123++ views; however, Trellis is generative and not strictly comparable and meaningful here. Other relevant pipelines such as MeshLRM is not mentioned or included for comparison.
3. Grid Resolution. DiMeR trains with 192^3 SDF grids "benefiting from our enhancement"; prior LRM-style baselines often default to 128^3 to the best of my knowledge. Please clarify whether your reported gains persist at equal grid resolution and whether CD improvements correlate with resolution. Ideally, a lower resolution variant should be included for completeness and would give a better understanding of the performance gain.
4. Normal FM dependency and failure modes: more qualitative failure cases and error-to-CD curves would provide more insights and help understand about the robustness of the model.

**Questions:**

1. Please add discussion on what DiMeR does that PRM cannot, and how your normal-only geometry branch differs from the recent "normal bridging" idea in Hi3DGen and mesh-LRM variants—ideally with side-by-side ablations where applicable.
2. Add discussions about resolution-control and sensitivity to normal FM outputs as mentioned in the weaknesses.

---

> ### Author Response · Authors · 2025-11-18
> **Response to Reviewer QJa6 (1/2)**
>
> We **sincerely** thank you for reviewing our paper and for your constructive comments. We have carefully addressed all the weaknesses and questions raised, providing additional experiments and detailed explanations. **The revisions have been highlighted in cyan in the updated manuscript.** We hope our responses adequately address your concerns, and we remain available for any further discussion or clarification you may require.
>
> ### **[W1] Novelty**
>
> We appreciate the reviewer’s careful positioning against contemporaries.
>
> Our work indeed builds on three active lines of research: (1) Feed-forward Mesh Reconstruction Models (e.g., LRM, InstantMesh), (2) geometry/texture disentanglement (e.g., CLAY), and (3) normal-based geometry cues (e.g., MeshLRM, InstantMesh, PRM). **What our DiMeR adds on top of these is a stronger inductive bias via strict modality-level *input* disentanglement for the elimination of ambiguity and hallucination in training (shown in Fig.2).** We exclusively utilize normal maps as the only input for the geometry reconstruction branch. To our knowledge, DiMeR is the first feed-forward mesh reconstruction framework that **purely conditions geometry on multi-view normals.** In ablations (Tab.3), switching the input to this normal-only geometry branch yields a substantial CD/F1 improvement and visibly reduces texture-induced geometric artifacts, indicating that this is not a minor engineering choice but **a key architectural contribution**. In addition, we make several architectural and training improvements within this disentangled framework.
>
> The focus of the previous works you mentioned is significantly different from ours. Concretely, **(1)** Fantasia3D is an SDS-based optimization framework that can only produce one 3D model for a text input using several hours. It doesn't involve a so-called input, and normal maps are only rendered as an auxiliary signal for SDS, not as the condition for geometry. **(2)** Clay’s input is still the RGB signal, which is proven to be ambiguous and results in hallucination in the training. **(3)** LRM-family models (LRM, InstantMesh, MeshLRM, MeshFormer, PRM) are RGB-based or RGB+normal-based encoders, where geometry remains entangled with appearance. **All these methods don’t consider the geometry’s ambiguity and hallucination produced by the RGB input.**
>
> ### **[W2] Fairness and more comparisons**
>
> Thanks for your suggestions. We acknowledge the reviewer's concern about comparing with Trellis, a generative model, in Tab.2. We have remove it to another discussion section. Moreover, we add the comparisons with MeshLRM based on GSO and OmniObject3D datasets in Tab.2 (**Sec. 4.2 noted in cyan of the revised paper**).
>
> |         | Chamfer Distance(↓) | F1-Score(↑) |
> |---------|:-------------------:|:-----------:|
> | MeshLRM |        0.079        |    0.933    |
> | DiMeR   |        0.052        |    0.981    |
>
>
> |         | Chamfer Distance(↓) | F1-Score(↑) |
> |---------|:-------------------:|:-----------:|
> | MeshLRM |        0.086        |    0.915    |
> | DiMeR   |        0.060        |    0.964    |
>
>
> Though Trellis is a generative model, we also supplement a fairer study from two aspects. **We have added the discussion in the Sec. 4.5 of the revised paper, noted in cyan color.**
>
> 1. We add the comparison for the Trellis based on the same multi-view input.
>
> |             | Chamfer Distance(↓) | F1-Score(↑) |
> |-------------|:-------------------:|:-----------:|
> | Trellis(MV) |        0.136        |    0.835    |
> | DiMeR(GT)   |        0.028        |    0.992    |
> | DiMeR(SN)   |        0.032        |    0.988    |
>
>
> 1. We add the comparison for the Trellis based on the output of Zero123plus.
>
> |                     | Chamfer Distance(↓) | F1-Score(↑) |
> |---------------------|:-------------------:|:-----------:|
> | Trellis original    |        0.119        |    0.859    |
> | Trellis+zero123plus |        0.155        |    0.783    |
> | DiMeR               |        0.052        |    0.981    |
>
>
> ### **[W3&Q2] Ablation study of the resolution**
>
> Thanks for the suggestion. We add the comparison experiment you mentioned at a 128 grid resolution and **update it to Sec. 4.4 of the revised paper (noted in cyan color)**. The experiment proves the improvement of our network architecture simplification. Furthermore, after reducing the resolution, our accuracy only decreased slightly and still remains higher than previous methods, InstantMesh and PRM.
>
> |        | Chamfer Distance(↓) | F1-Score(↑) |
> |--------|:-------------------:|:-----------:|
> | 128res |        0.032        |    0.987    |
> | 192res |        0.028        |    0.992    |

---

> ### Author Response · Authors · 2025-11-18
> **Response to Reviewer QJa6 (2/2)**
>
> ### **[W4&Q2] Normal Foundation Model (FM) dependency and failure modes**
>
> Thanks for your suggestion. We add the experiments based on different normal foundation models as DiMeR’s input and **update it to Sec. 4.5 of the revised paper (noted in cyan color)**. DiMeR empirically shows strong robustness across different normal prediction models. Furthermore, we can observe that the CD value exhibits a decreasing trend as the normal prediction error decreases.
>
> On the GSO dataset, the results are
>
> |                    | Chamfer Distance(↓) | F1-Score(↑) | Normal Prediction Mean Error(↓) |
> |--------------------|:-------------------:|:-----------:|:-------------------------------:|
> | GeoWizard+DiMeR    |        0.039        |    0.982    |              17.673             |
> | MariGold+DiMeR     |        0.037        |    0.986    |              17.400             |
> | DSINE+DiMeR        |        0.040        |    0.978    |              17.953             |
> | Lotus-G+DiMeR      |        0.033        |    0.988    |              17.151             |
> | Lotus-D+DiMeR      |        0.035        |    0.987    |              16.606             |
> | StableNormal+DiMeR |        0.032        |    0.988    |              16.818             |
>
>
> On the OmniObject3D, the results are
>
> |                    | Chamfer Distance(↓) | F1-Score(↑) | Normal Prediction Mean Error(↓) |
> |--------------------|:-------------------:|:-----------:|:-------------------------------:|
> | GeoWizard+DiMeR    |        0.037        |    0.987    |              23.129             |
> | MariGold+DiMeR     |        0.038        |    0.984    |              22.934             |
> | DSINE+DiMeR        |        0.046        |    0.974    |              23.010             |
> | Lotus-G+DiMeR      |        0.036        |    0.988    |              21.523             |
> | Lotus-D+DiMeR      |        0.034        |    0.989    |              21.065             |
> | StableNormal+DiMeR |        0.030        |    0.993    |              21.205             |
>
>
> We also add a failure mode when inputting the scene-level sparse-view normal maps **at the last of the appendix**. Since we train DiMeR on the object-wise normal maps with a white background, we can not accept the scene-level sparse views as input.
>
> ### **[Q1] Difference from PRM and Hi3DGen**
>
> **Difference from PRM.** PRM focuses on using PBR to enhance the reconstruction quality under complex lighting environments, and its input is also RGB images. We emphasize that we use normal-only input for geometry reconstruction and use RGB for texture.
>
> **Difference from Hi3DGen.** Our model focuses on feed-forward reconstruction models. At the same time, Hi3DGen applies similar concepts to diffusion models. **The key difference** lies in: we reconstruct geometry using multi-view normal maps, whereas Hi3DGen generates geometry solely from a single-view normal map. Generating an object from a single-view normal map constitutes an underconstrained problem, as relying on a single normal map leaves vast unobserved regions. In contrast, our method employs sparse-view normal maps, enabling us to capture the majority of an object's surface details.
>
> We demonstrate side-by-side ablation studies for the evolution of input and output in Tab.3. The term “RGB+Normal→Geometry+Texture” is similar to the MeshLRM series model with normal as additional input. And when we discard the output of texture in a single backbone, we achieve a slight improvement. Then, we discard the RGB input, and we achieve a huge enhancement.

---

> ### Author Response · Authors · 2025-11-27
>
> Dear Reviewer QJa6,
>
> Thank you once again for your valuable feedback. We have conducted additional experiments and made revisions to the paper based on your suggestions. **As the discussion phase is nearing its conclusion**, we would like to know if our responses have addressed your concerns. We are looking forward to hearing from you.
>
> Best, Authors

---

### Official Review · Reviewer_68mg · 2025-11-02

**Soundness:** 3
**Presentation:** 3
**Contribution:** 2
**Rating:** 4
**Confidence:** 4

**Summary:**

The paper presents a disentangled dual-branch model for 3D mesh reconstruction that separately learns geometry and texture. The geometry branch predicts shape using only normal maps, while the texture branch learns appearance from RGB images. This separation reduces ambiguity in cases where textures might hide geometric errors. This paper also improves mesh extraction with a simplified version of FlexiCubes, combined with 3D supervision and PBR-based regularization. The method achieves better results over existing approaches such as PRM and InstantMesh on both quantitative and qualitative metrics across sparse-view, single-image, and text-to-3D reconstruction tasks.

**Strengths:**

* The motivation of the paper is clear. The paper is well-written and the proposed method is easy to follow and the overall presentation is clear.

* The paper is solving a relevant problem. Geometry ambiguity introduced due to texture is a common problem in optimization based 3D reconstruction pipelines.

* Comprehensive evaluation is performed over several tasks.

**Weaknesses:**

* Novelty: While the idea of learning disentangled representations for geometry and texture using different input modalities (normals for geometry and RGB images for texture) is sound, the overall approach of using normal supervision to regularize reconstruction is very standard with already known components.

* Experimental Results: Overall, the results presented in both the main paper and the supplementary material appear relatively simple compared to the quality achieved by recent state-of-the-art mesh reconstruction methods.

* No analysis and discussions on failure cases and limitations has been done in the paper.

**Questions:**

Please follow the weakness section.

* Why is Trellis omitted from the comparison in the sparse-view reconstruction task in Table 1? Since Trellis is capable of predicting meshes from sparse-view images, it seems reasonable to include it for a fair comparison.

* In Table 2, why is only a single image provided as input to methods like Trellis, while DiMeR receives sparse-view inputs generated from Zero-123? Wouldn’t it be more consistent and fair to use the same input setup for all methods?

---

> ### Author Response · Authors · 2025-11-18
> **Response to Reviewer 68mg (1/2)**
>
> We **sincerely** thank you for reviewing our paper and for your constructive comments. We have carefully addressed all the weaknesses and questions raised, providing additional experiments and detailed explanations. **The revisions have been highlighted in cyan in the updated manuscript.** We hope our responses adequately address your concerns, and we remain available for any further discussion or clarification you may require.
>
> ### **[W1] Novelty**
>
> Thank you for your recognition of our idea (only normal for geometry and RGB for texture). Using normal supervision is introduced by InstantMesh/MeshLRM/PRM, and it has been a common component in the feed-forward mesh reconstruction models, which is not the contribution of our paper.
>
> **What our DiMeR adds on top of these is a stronger inductive bias via strict modality-level *input* disentanglement for the elimination of ambiguity and hallucination in training (shown in Fig.2).**  The key insight of our paper is that **using normal-only inputs for geometry reconstruction** can significantly enhance geometric accuracy. As shown in Fig.2 of the main paper, we argue that RGB input may result in ambiguity and training objective conflict. To solve this problem, we propose to utilize the unbiased modality, the normal maps from multiple views, as the only input to predict the geometry. In addition, we make several architectural and training improvements within this disentangled framework.
>
> ### **[W2] Performance comparison with over-sized large model**
>
> We agree that, in terms of pure visual quality, our results cannot yet compete with very recent large-scale models such as Hunyuan 3D, Tripo, and similar systems. These methods typically rely on **very large DiT backbones (over 20B) and massive training datasets**, together with substantial computational budgets. In contrast, the primary goal of our work is **not** to push the absolute SOTA in terms of “best-looking” renderings at any cost, but rather to **propose and validate our core contribution**: a strictly disentangled geometry–texture design with normal-only geometry inputs that significantly improves reconstruction accuracy and robustness.
> Therefore, we position DiMeR as a **lightweight feed-forward reconstruction model (only 0.4B, ~2% of the large models)**, and we mainly compare against methods of a similar type and scale. **Under this setting**, our approach consistently brings **clear and significant improvements** over prior feed-forward mesh reconstruction baselines. Importantly, our design is largely **orthogonal to model size and training scale**: in principle, the same modality-level disentanglement and normal-only geometry input can be incorporated into larger DiT-based architectures.
> We expect that scaling up DiMeR with larger backbones and data would naturally yield more visually impressive results, while preserving the benefits demonstrated in our current experiments.

---

> ### Author Response · Authors · 2025-11-18
> **Response to Reviewer 68mg (2/2)**
>
> ### **[Q1] Comparison with multi-view Trellis in Tab.1**
>
> Thanks for your suggestions. **We add the discussion in Sec. 4.5 of the revised paper.** We have added the relevant comparative experiments with a multi-view variant of Trellis in the sparse-view reconstruction setting. The results are summarized below: DiMeR achieves significantly higher reconstruction accuracy at a notably lower computational cost compared with Trellis.
>
> |             | Chamfer Distance(↓) | F1-Score(↑) |
> |-------------|:-------------------:|:-----------:|
> | Trellis(MV) |        0.136        |    0.835    |
> | DiMeR(GT)   |        0.028        |    0.992    |
> | DiMeR(SN)   |        0.032        |    0.988    |
>
> Even the multi-view version of the Trellis is still not a reconstruction model. Trellis is primarily designed as a generative model. Specifically, Trellis’s multi-view input strategy involves randomly sampling viewpoints during different denoising steps.  Only one randomly selected view is received at each denoising step. This cannot aggregate all views simultaneously, which may limit its ability to fully exploit multi-view consistency in our reconstruction setting. Moreover, Trellis can't generate 3D meshes that perfectly match the input images. These points explain its lower reconstruction accuracy under our evaluation protocol.
>
> ### **[Q2] Fairness about Table 2**
>
> Thanks for your suggestions. DiMeR is fundamentally a sparse-view-to-3D model. Similar to prior works such as InstantMesh and PRM, it can also be combined with a multi-view diffusion model to perform **single-image-to-3D** reconstruction. Our experiments in Table 2 follow this commonly used pipeline, based on the same single-image input.
>
> In Table 2, trellis receives the original single view, since it is a single-image-to-3D model. CRM trained a multi-view diffusion model additionally with CCM output, so we follow its setting and use the same single-view input. For the other methods, they all apply the single-view-zero123plus-multi-view-to-3D pipeline. We also input the same single-view into the zero123plus model.
>
> To eliminate ambiguity, we conduct the experiments based on the sparse-view Trellis version with the output of zero123plus, i.e., single-view-zero123plus-multi-view-Trellis-3D. **Moreover, we add the discussion in Sec. 4.5 of the revised paper.** As discussed in the last question, Trellis is still not a reconstruction model that can really get multi-view information. Therefore, its performance is limited, even worse than original single-view input.
>
> |                     | Chamfer Distance(↓) | F1-Score(↑) |
> |---------------------|:-------------------:|:-----------:|
> | Trellis original    |        0.119        |    0.859    |
> | Trellis+zero123plus |        0.155        |    0.783    |
> | DiMeR               |        0.052        |    0.981    |
>
> ### **[W3] Limitations and failure cases**
>
> Thanks for your suggestions. **We add the relative part at the Sec.5 and the last of the appendix in the revised paper.** The main limitations of our work are 1) Since we train DiMeR on the object-wise normal maps with a white background, we can not accept the scene-level sparse views as input. 2) On the single-image-to-3D task, the enhancement of the 2.5D diffusion model, including resolution and accuracy, would help DiMeR achieve better quality.
>
> Moreover, we also add a failure mode when inputting the scene-level sparse-view normal maps **at the last of the appendix**. Since we train DiMeR on the object-wise normal maps with a white background, we can not accept the scene-level sparse views as input.

---

> ### Author Response · Authors · 2025-11-27
>
> Dear Reviewer 68mg,
>
> Thank you once again for your valuable feedback. We have conducted additional experiments and made revisions to the paper based on your suggestions. **As the discussion phase is nearing its conclusion**, we would like to know if our responses have addressed your concerns. We are looking forward to hearing from you.
>
> Best, Authors

---

### Author Response · Authors · 2025-12-02
**Rebuttal Summary of Submission 1623**

Dear Area Chair,

We would like to start by expressing our deepest gratitude for your exceptional handling of our submission. We sincerely thank you for your hard work during such a challenging time.

### 1. Summary of Strengths Acknowledged by Reviewers

- **Clear motivation and presentation.** All reviewers recognized that our paper is well-written with clear motivation. (68mg, QJa6, jDrM, SAKn)
- **Clean and effective disentanglement design.** The proposed normal-only input for geometry reconstruction is recognized as a sound and effective approach to eliminate ambiguity (jDrM, SAKn).
- **Promising, substantial, and strong improvements.** Extensive experiments on GSO and OmniObject3D show substantial improvements over prior methods. (QJa6, jDrM, SAKn)

### 2. Summary of Additional Experiments and Revisions

In response to **all** the concerns from reviewers, we have conducted extensive additional experiments:

- Fairness
    - Added MeshLRM comparison
    - Added multi-view Trellis comparison
    - Trellis+Zero123plus pipeline comparison
- Ablation
    - Grid resolution study (128 vs 192) showing improvements persist at equal resolution
- Robustness for different base models
    - Tested 6 different normal foundation models (GeoWizard, MariGold, DSINE, Lotus-G, Lotus-D, StableNormal) based on our DiMeR
    - Zero123plus joint normal experiments
- Noise Sensitivity
    - Gaussian noise perturbation experiments (σ=0.0~0.5) demonstrating graceful degradation
- Limitations
    - Added failure cases and limitations discussion in Sec. 5 and Appendix

**All revisions have been highlighted in cyan in the updated manuscript.** These experiments can solve reviewers’ questions and concerns.

Moreover, we also provide more clarification and discussion about novelty, differences from prior works, and some specific design choices.

Thank you once again for your selfless dedication and for ensuring a rigorous review process.

Sincerely,

The Authors

---

### Meta-Review · Area_Chair_KV76 · 2026-01-02

**Summary:**

This paper receives mixed ratings: 2 marginally below the acceptance threshold, 2 marginally above the acceptance threshold. Reviewers concerns are centered around following points:
1. novelty is limited.
2. experiments are not comprehensive enough, such as missing baseline (MeshLRM), and sensitivity to noisy normal maps.
3. lack of analysis on limitation and failure cases.

AC have carefully checked reviewers' comments and authors' responses. Reviewers didn't participate in the discussion period. AC found all concerns have been well addressed with supporting results.

In terms of novelty, as authors stated, the main novelty is a stronger inductive bias via strict modality-level input disentanglement for the elimination of ambiguity and hallucination in training (shown in Fig.2), which means using normal-only inputs for geometry reconstruction. This is a simple yet very effective insight, supported by results across datasets, tasks and different estimated normal maps.

In terms experiments, all required comparisons and analysis are provided in the responses.

Finally, limitation and failure cases are also discussed in the responses.

Therefore, AC believes this submission is a valuable addition to ICLR. The decision is accept.

**Reviewer Concerns:**

Reviewers mainly concern on novelty, additional experiments and discussion of limitation and failure cases. All of these concerns are well addressed by authors' responses.

**Reviewer Scores:**

All reviewers will raise their scores if they check the responses and participate in the discussion.

---

### Decision · Program_Chairs · 2026-01-26

Accept (Poster)